# $L_2$-Uniform Stability of Randomized Learning Algorithms: Sharper Generalization Bounds and Confidence Boosting

**Xiao-Tong Yuan**
School of Intelligence Science and Technology
Nanjing University
Suzhou, 215163, China
xtyuan1980@gmail.com

**Ping Li**
VecML Inc. www.vecml.com
Bellevue, WA 98004, USA
pingli98@gmail.com

## Abstract

Exponential generalization bounds with near-optimal rates have recently been established for uniformly stable algorithms (Feldman and Vondrák, 2019; Bousquet et al., 2020). We seek to extend these best known high probability bounds from deterministic learning algorithms to the regime of randomized learning. One simple approach for achieving this goal is to define the stability for the expectation over the algorithm's randomness, which may result in sharper parameter but only leads to guarantees regarding the on-average generalization error. Another natural option is to consider the stability conditioned on the algorithm's randomness, which is way more stringent but may lead to generalization with high probability jointly over the randomness of sample and algorithm. The present paper addresses such a tension between these two alternatives and makes progress towards relaxing it inside a classic framework of confidence-boosting. To this end, we first introduce a novel concept of $L_2$-uniform stability that holds uniformly over data but in second-moment over the algorithm's randomness. Then as a core contribution of this work, we prove a strong exponential bound on the first-moment of generalization error under the notion of $L_2$-uniform stability. As an interesting consequence of the bound, we show that a bagging-based meta algorithm leads to near-optimal generalization with high probability jointly over the randomness of data and algorithm. We further substantialize these generic results to stochastic gradient descent (SGD) to derive sharper exponential bounds for convex or non-convex optimization with natural time-decaying learning rates, which have not been possible to prove with the existing stability-based generalization guarantees.

## 1 Introduction

In many statistical learning problems, we are interested in designing a randomized algorithm $A : \mathcal{Z}^N \times \mathcal{R} \mapsto \mathcal{W}$ that maps a training data sample $S = \{Z_i\}_{i \in [N]} \in \mathcal{Z}^N$ with an algorithm's random parameter $\xi \in \mathcal{R}$ to a model $A(S, \xi) \in \mathcal{W}$. Here $\mathcal{Z}$ and $\mathcal{R}$ are some measurable sets, and $\mathcal{W}$ is a closed subset of an Euclidean space. The ultimate goal is to find a suitable algorithm such that the following population risk evaluated at the model should be as small as possible:

$$R(A(S, \xi)) := \mathbb{E}_Z[\ell(A(S, \xi); Z)],$$

where $Z \in \mathcal{Z}$ and $\ell : \mathcal{W} \times \mathcal{Z} \mapsto \mathbb{R}^+$ is a non-negative bounded loss function whose value $\ell(w; z)$ measures the loss evaluated at $z$ with parameter $w$. It is generally the case that the underlying data distribution is unknown, and in this case the data points $Z_i$ are usually assumed to be independent.

37th Conference on Neural Information Processing Systems (NeurIPS 2023).

Then, a natural alternative measurement that mimics the computationally intractable population risk is the empirical risk given by

$$R_S(A(S,\xi)) := \mathbb{E}_{Z \sim \mathtt{Unif}(S)}[\ell(A(S,\xi);Z)] = \frac{1}{N}\sum_{i=1}^{N}\ell(A(S,\xi);Z_i).$$

The bound on the difference between the population and empirical risks is of central interest in understanding the generalization performance of a learning algorithm. In particular, we hope to derive a suitable law of large numbers, i.e., a sample size vanishing rate $b_N$ such that the generalization bound $|R_S(A(S,\xi)) - R(A(S,\xi))| \lesssim b_N$ holds with high probability over the randomness of $S$ and hopefully the randomness of $\xi$ as well. Let $R^* := \min_{w \in \mathcal{W}} R(w)$ be the optimal value of the population risk. Conditioned on $S$, suppose that $A(S,\xi)$ is an almost minimizer of the empirical risk $R_S$ such that $R_S(A(S,\xi)) - \min_{w \in \mathcal{W}} R_S(w) \le \varepsilon$, then the generalization bound immediately implies an *excess risk* bound $R(A(S,\xi)) - R^* \lesssim b_N + \frac{1}{\sqrt{N}} + \varepsilon$ based on the standard risk decomposition and Hoeffding's inequality. Therefore, generalization guarantees also play a crucial role in understanding the stochastic optimization performance of a learning algorithm.

A powerful proxy for analyzing the generalization bounds is the *stability* of learning algorithms to changes in the training dataset. Since the seminal work of Bousquet and Elisseeff (2002), stability has been extensively demonstrated to beget dimension-independent generalization bounds for deterministic learning algorithms (Mukherjee et al., 2006; Shalev-Shwartz et al., 2010), as well as for randomized learning algorithms such as bagging and SGD (Elisseeff et al., 2005; Hardt et al., 2016). So far, the best known results about generalization bounds are offered by approaches based on the notion of uniform stability (Feldman and Vondrák, 2018, 2019; Bousquet et al., 2020; Klochkov and Zhivotovskiy, 2021) which is independent to the underlying distribution of data. For randomized algorithms, the definition of uniform stability can be extended in two natural ways by respectively considering 1) the stability averaged over the algorithm's randomness (Hardt et al., 2016) and 2) the stability conditioned on the algorithm's randomness (Feldman and Vondrák, 2019). The former is simpler to show but typically leads to on-average generalization bounds, while the latter is relatively more stringent but may yield deviation bounds given that the conditional stability holds with high probability over the algorithm's randomness. Between these two extreme cases, however, the generalization behavior of randomized learning algorithm still remains largely under explored.

To address the above mentioned theoretical gap between the current lines of results, we explore the opportunities of *deriving exponential generalization bounds for randomized learning algorithms beyond the notions of on-average stability and conditional stability*. A concrete working example of our study is the widely used stochastic gradient descent (SGD) algorithm that carries out the following recursion for all $t \ge 1$ with learning rate $\eta_t > 0$:

$$w_t := \Pi_{\mathcal{W}}\left(w_{t-1} - \eta_t \nabla_w \ell(w_{t-1}; Z_{i_t})\right), \tag{1}$$

where $i_t \in [N]$ is a random index of data under with or without replacement sampling, and $\Pi_{\mathcal{W}}$ is the Euclidean projection operator associated with $\mathcal{W}$. The in-expectation generalization of SGD has been studied under on-average stability (Hardt et al., 2016; Zhou et al., 2022; Lei and Ying, 2020), while the exponential bounds have recently been established given that the stability holds with high probability over the sampling path of SGD (Feldman and Vondrák, 2019; Bassily et al., 2020).

## 1.1 Prior results

Let us start by briefly reviewing some state-of-the-art exponential generalization bounds under the notion of uniform stability and its randomized variants. We denote by $S \doteq \tilde{S}$ if a pair of data sets $S$ and $\tilde{S}$ differ in a single element. A randomized learning algorithm $A$ is said to have *on-average* $\gamma_N$-*uniform stability* (Elisseeff et al., 2005) if it satisfies the following uniform bound:

$$\sup_{S \doteq \tilde{S}, Z \in \mathcal{Z}}\left|\mathbb{E}_\xi\left[\ell(A(S,\xi);Z) - \ell(A(\tilde{S},\xi);Z)\right]\right| \le \gamma_N. \tag{2}$$

This definition is equivalent to the concept of uniform stability defined for the expectation of loss $\mathbb{E}_\xi[\ell(A(S,\xi);Z)]$. Suppose that the loss function is bounded in the interval $[0, M]$. Then essentially it has been shown in Feldman and Vondrák (2019) that for any $\delta \in (0,1)$, with probability at least $1 - \delta$ over $S$, the on-average generalization error is upper bounded by

$$|\mathbb{E}_\xi[R(A(S,\xi)) - R_S(A(S,\xi))]| \lesssim \gamma_N \log(N)\log\left(\frac{N}{\delta}\right) + M\sqrt{\frac{\log(1/\delta)}{N}}. \tag{3}$$

Bousquet et al. (2020) later derived a slightly improved exponential bound that implies

$$|\mathbb{E}_\xi \left[R(A(S,\xi)) - R_S(A(S,\xi))\right]| \lesssim \gamma_N \log(N) \log\left(\frac{1}{\delta}\right) + M\sqrt{\frac{\log(1/\delta)}{N}}. \tag{4}$$

These bounds are near-tight (up to logarithmic factors) in the sense of an $\mathcal{O}\left(\gamma_N \log\left(\frac{1}{\delta}\right) + \sqrt{\frac{\log(1/\delta)}{N}}\right)$ lower deviation bound on sum of random functions with $\gamma_N$-uniform stability (Bousquet et al., 2020, Proposition 9). Concerning the excess risk bound, Klochkov and Zhivotovskiy (2021) essentially derived the following result using the sample-splitting techniques of Bousquet et al. (2020):

$$\mathbb{E}_\xi \left[R(A(S,\xi))\right] - R^* \lesssim \Delta_{\text{opt}} + \mathbb{E}[\Delta_{\text{opt}}] + \gamma_N \log(N) \log\left(\frac{1}{\delta}\right) + \frac{(M+B)\log(1/\delta)}{N}, \tag{5}$$

where $\Delta_{\text{opt}} := \mathbb{E}_\xi \left[R_S(A(S,\xi))\right] - \min_{w \in \mathcal{W}} R_S(w)$ represents the in-expectation empirical risk sub-optimality, and $B$ is the constant of the generalized Bernstein condition (Koltchinskii, 2006). While sharp in the dependence on sample size, one common limitation of the above uniform stability implied generalization and risk bounds lies in that these high-probability results only hold *in expectation* with respect to $\xi$, the internal randomness of algorithm.

Alternatively, consider that $A$ has $\gamma_N$-uniform stability with probability at least $1 - \delta'$ for some $\delta' \in (0,1)$ over the random draw of $\xi$, i.e.,

$$\mathbb{P}\left\{\sup_{S \doteq \tilde{S}, Z \in \mathcal{Z}} |\ell(A(S,\xi);Z) - \ell(A(\tilde{S},\xi);Z)| \le \gamma_N\right\} \ge 1 - \delta'. \tag{6}$$

Suppose that the randomness of $A$ is independent of the training set $S$. Then the bound of Bousquet et al. (2020) naturally implies that with probability at least $1 - \delta - \delta'$ over $S$ and $\xi$,

$$|R(A(S,\xi)) - R_S(A(S,\xi))| \lesssim \gamma_N \log(N) \log\left(\frac{1}{\delta}\right) + M\sqrt{\frac{\log(1/\delta)}{N}}. \tag{7}$$

This is by far the best known generalization bound of randomized stable algorithms that hold with high probability jointly over the randomness of data and algorithm. The result, however, relies heavily on the high-probability uniform stability as expressed in (6). For the SGD recursion (1) with fixed learning rate $\eta_t \equiv \eta$, it is possible to show that $\gamma_N \lesssim \eta\sqrt{T} + \frac{\eta T}{N}$ and $\delta' = N\exp(-\frac{N}{2})$ in (6) (Bassily et al., 2020). For SGD with time decaying learning rates, which has been widely studied in theory (Harvey et al., 2019; Rakhlin et al., 2012) and applied in practice for training popular deep nets such as ResNet and DenseNet (Bengio et al., 2017), it is not clear if the condition in (6) is still valid for $\gamma_N$ and $\delta'$ of interest. Madden et al. (2020) indeed have established a high-probability uniform stability bound for minibatch SGD with learning rates $\eta_t \lesssim \frac{1}{Nt}$. However, such a fairly conservative choice of learning rates tends to impair the empirical minimization performance of SGD and thus is of limited interest from the perspective of risk minimization.

More specially for randomized learning methods such as bagging (Breiman, 1996) and SGD, the randomness of algorithm can be precisely characterized by a vector of i.i.d. parameters $\xi = \{i_1, ..., i_t\}$ which are independent on data $S$. In such cases, assume additionally that $A(S,\xi)$ has uniform stability with respect to $\xi$ conditioned on $S$, i.e., $\sup_{\xi \doteq \tilde{\xi}} |\ell(A(S,\xi)) - \ell(A(S,\tilde{\xi}))| \le \rho_T$. Then the following exponential bound has been derived by Elisseeff et al. (2005):

$$|R(A(S)) - R_S(A(S))| \lesssim \gamma_N + \left(\frac{1 + N\gamma_N}{\sqrt{N}} + \sqrt{T}\rho_T\right)\sqrt{\log\left(\frac{1}{\delta}\right)}. \tag{8}$$

Provided that $\gamma_N \lesssim \frac{1}{N}$ and $\rho_T \lesssim \frac{1}{T}$, the above bound shows that the generalization bound scales as $\mathcal{O}\left(\frac{1}{\sqrt{N}} + \frac{1}{\sqrt{T}}\right)$ with high probability. However, the rate of the above bound is sub-optimal and will show no guarantee on convergence if $\gamma_N \gtrsim \frac{1}{\sqrt{N}}$ and/or $\rho_T \gtrsim \frac{1}{\sqrt{T}}$. As an example, for non-convex SGD with learning rate $\eta_t = O\left(\frac{1}{t}\right)$, it can be shown that $\gamma_N \lesssim \frac{\sqrt{T}}{N}$ and $\rho_T$ scales as large as $\mathcal{O}(1)$.

**Open problem.** So far, it still remains open if the exponential generalization bounds for deterministic uniformly stable algorithms might be extended to randomized learning algorithms under the variants of uniform stability tighter than the on-average version (2) but less restrictive than the high-probability

version (6). Particularly, we are interested in the following notion of $L_2$-*uniform stability* (as formally introduced in Definition 1) with parameter $\gamma_{\mathrm{L}_2,N}$:

$$\sup_{S \doteq \tilde{S}, Z \in \mathcal{Z}} \mathbb{E}_\xi \left[ \left( \ell(A(S,\xi);Z) - \ell(A(\tilde{S},\xi);Z) \right)^2 \right] \leq \gamma_{\mathrm{L}_2,N}^2, \tag{9}$$

which represents a second-moment variant of the uniform stability for randomized learning algorithms. For example, as we will shortly show in Section 4 that SGD with practical time-decaying learning rates has $L_2$-uniform stability with favorable parameters. The main goal of the present work is to derive sharper exponential generalization bounds for randomized learning algorithms under the notion of $L_2$-uniform stability.

### 1.2 Overview of our contribution

The fundamental contribution of this work is a near-optimal first-moment generalization error bound for $L_2$-uniformly stable algorithms, which is summarized in Theorem 1 and highlighted below:

$$\mathbb{E}_\xi \left[ |R(A(S,\xi)) - R_S(A(S,\xi))| \right] \lesssim \gamma_{\mathrm{L}_2,N} \log(N) \log\left(\frac{1}{\delta}\right) + M\sqrt{\frac{\log(1/\delta)}{N}}.$$

While our first-moment bound above has an identical convergence rate to that of the on-average bound in (4), the former is stronger in the sense that the expectation is taken outside the generalization gap and thus implies the latter where the expectation is taken inside. The key ingredients of our analysis are a set of fine-grained concentration inequalities for randomized functions (Proposition 1) and sums of randomized functions (Proposition 2), which respectively generalize the classic bounded-difference inequalities and a prior result of Bousquet et al. (2020) under the considered $L_2$-uniform bounded difference conditions. These generalized concentration inequalities and their proof arguments are novel to our knowledge and should be of independent interests in analyzing randomized functions.

As an important consequence of our main result, we reveal that a bagging-based meta procedure (see Algorithm 1) can be used to boost the confidence of generalization for $L_2$-uniformly stable algorithms. More specifically, in the presented bagging procedure we independently run a randomized algorithm $A$ multiple $K$ times over a fraction of the training set to obtain $K$ solutions. Then we evaluate the validation error of these candidate solutions over a holdout training subset, and output the solution that has the smallest training-validation gap. Our result in Theorem 2 shows that for any confidence level $\delta \in (0,1)$, setting $K \asymp \log(\frac{1}{\delta})$ yields a near-optimal generalization bound for the selected solution that holds with high probability jointly over the randomness of data and algorithm.

We have substantialized our results to SGD with smooth (Corollary 1) or non-smooth (Corollary 2) convex losses, and smooth non-convex losses (Corollary 3) as well. For an instance, our result in Corollary 1 shows that when invoked to SGD with smooth convex loss and learning rates $\eta_t = \mathcal{O}(\frac{1}{\sqrt{t}})$, the generalization bound of the output of Algorithm 1 may scale as $\mathcal{O}\left( \log(N) \log\left(\frac{1}{\delta}\right) \sqrt{\frac{\log(T)}{N}} + \frac{\sqrt{T}}{N} \right)$. To compare with the $\mathcal{O}(\frac{\sqrt{T}}{N})$ in-expectation bound of smooth convex SGD (Hardt et al., 2016), our bound above for the boosted SGD is comparable in convergence rate while it holds with high probability jointly over the randomness of data and sampling path.

## 2 $L_2$-Uniform Stability and Generalization

### 2.1 Notation and definitions

Let us introduce some notation to be used in our analysis. We abbreviate $[N] := \{1, ..., N\}$. Recall that $S = \{Z_i\}_{i \in [N]}$ is a set of i.i.d. training data. Denote by $S' = \{Z_i'\}_{i \in [N]}$ an independent copy of $S$ and we write $S^{(i)} = \{Z_1, ..., Z_{i-1}, Z_i', Z_{i+1}, ..., Z_N\}$. For a real-valued random variable $Y$, its $L_q$-norm for $q \geq 1$ is given by $\|Y\|_q = (\mathbb{E}[|Y|^q])^{1/q}$. By definition it can be verified that $\forall q \geq 2$,

$$\|Y\|_q^2 = (\mathbb{E}[|Y|^q])^{2/q} = \left( \mathbb{E}[|Y^2|^{q/2}] \right)^{2/q} = \left\| Y^2 \right\|_{q/2}. \tag{10}$$

Let $h : \mathcal{Z}^N \mapsto \mathbb{R}$ be some measurable function and consider the random variable $h(S) = h(Z_1, ..., Z_N)$. For $h(S)$ and any index set $I \subseteq [N]$, we define the following abbreviations:

$$h(S_I) := \mathbb{E}\left[ h(S) \mid S_I \right], \quad \|h\|_q(S_I) := \left( \mathbb{E}\left[ |h(S)|^q \mid S_I \right] \right)^{1/q}.$$

We say a function $f$ to be $G$-Lipschitz continuous over $\mathcal{W}$ if $|f(w) - f(\tilde{w})| \leq G\|w - \tilde{w}\|$ for all $w, \tilde{w} \in \mathcal{W}$, and it is $L$-smooth if $\|\nabla f(w) - \nabla f(\tilde{w})\| \leq L\|w - \tilde{w}\|$. For a pair of functions $f, f' \geq 0$, we use $f \lesssim f'$ (or $f' \gtrsim f$) to denote $f \leq cf'$ for some universal constant $c > 0$.

In the following definition, we formally introduce the concept of $L_2$-uniform stability for randomized learning algorithms to be investigated in this work.

**Definition 1** ($L_2$-Uniform stability of randomized learning algorithms). *We say a randomized learning algorithm $A : \mathcal{Z}^N \times \mathcal{R} \mapsto \mathcal{W}$ to have $L_2$-uniform stability with parameter $\gamma_{L_2,N} \geq 0$ if*

$$\sup_{S, Z_i', Z} \mathbb{E}_\xi \left[ \left( \ell(A(S, \xi); Z) - \ell(A(S^{(i)}, \xi); Z) \right)^2 \right] \leq \gamma_{L_2,N}^2.$$

**Remark 1.** *By definition the $L_2$-uniform stability has a second-moment dependence on the internal randomness of algorithm conditioned on data, while it is invariant to the data distribution. This justifies the name of such a notion of mixed algorithmic stability.*

**Remark 2.** *On one hand, by Jensen's inequality the $L_2$-uniform stability implies the on-average uniform stability defined in (2). On the other hand, the second-order form of $L_2$-uniform stability is by definition weaker than the high-probability uniform stability in (6). If the algorithm's randomness $\xi$ can be expressed as a set of i.i.d. random bits, then the $L_2$-uniform stability is also weaker than the conditional uniform stability conditioned on data $S$ (Elisseeff et al., 2005).*

Throughout this paper, we assume for simplicity that the output models $A(S^{(i)}, \xi)$ and $A(S, \xi)$ share the same internal random bit $\xi$ which is invariant to data. With similar analysis techniques, it is indeed possible to extend Definition 1 and our main results to the general setting where the randomness of algorithm is allowed to be dependent on data, such as in posterior sampling for Bayesian learning.

## 2.2 Concentration inequalities for randomized functions

We begin by establishing in the following result a group of first- and second-order concentration inequalities (in moments) for *randomized* functions of independent random variables.

**Proposition 1.** *Let $S = \{Z_1, Z_2, ..., Z_N\}$ be a set of independent random variables valued in $\mathcal{Z}$ and $\xi$ be a random variable valued in $\mathcal{R}$. Let $g : \mathcal{Z}^N \times \mathcal{R} \mapsto \mathbb{R}$ be a measurable function that satisfies the following $L_2$-bounded-difference condition:*

$$\sup_{S, Z_i'} \mathbb{E}_\xi \left[ \left( g(S, \xi) - g(S^{(i)}, \xi) \right)^2 \right] \leq \beta^2.$$

*Then for any $q \geq 2$,*

$$\left\| \mathbb{E}_\xi \left[ |g(S, \xi) - \mathbb{E}_S [g(S, \xi)]| \right] \right\|_q \leq 3\beta\sqrt{Nq}, \tag{11}$$

*and*

$$\left\| \mathbb{E}_\xi \left[ (g(S, \xi) - \mathbb{E}_S [g(S, \xi)])^2 \right] \right\|_q \leq 68N\beta^2 q. \tag{12}$$

*Proof in sketch.* Let us consider $h(S) := \mathbb{E}_\xi \left[ |g(S, \xi) - \mathbb{E}_S[g(S, \xi)]| \right]$. The given $L_2$-bounded-difference condition implies that $h(S)$ has bounded-difference property. Then the desired first-order bound in (11) can be obtained by respectively invoking a moment Efron-Stein inequality (Boucheron et al., 2005, Theorem 2) to upper bound $\|h(S) - \mathbb{E}[h(S)]\|_q$ and a slightly modified Efron-Stein inequality to bound the mean $\mathbb{E}[h(S)]$. To prove the second-order concentration bound, we consider the function $h'(S) := \mathbb{E}_\xi \left[ (g(S, \xi) - \mathbb{E}_S[g(S, \xi)])^2 \right]$, which can be shown to be *weakly self-bounding* (see Definition 2) under the $L_2$-bounded-difference condition. Then the desired bound (12) can be derived by applying the upper tail bound of Boucheron et al. (2005, Theorem 6.19) and lower tail bound of Klochkov and Zhivotovskiy (2021, Proposition 3.1) for weakly self-bounding functions. See Appendix A.2 for a detailed proof of this result. □

The moment bound in (11) extends the McDiarmid's (bounded difference) inequality (McDiarmid et al., 1989) to randomized functions with the $L_2$-bounded-difference property. The second-order concentration bound in (12) is crucial for proving the moment bound of sums in Proposition 2, as it can be used to sharply control some second-order components involved in the arguments. These generic inequalities are expected to be of independent interests for understanding the first-/second-order concentration behavior of randomized functions.

## 2.3 A moment inequality for sums of randomized functions

As a key intermediate result, we further establish in the following proposition a moment concentration inequality for sums of randomized functions that satisfy the $L_2$-bounded-difference condition. This result extends the moment bound for sums of functions (Bousquet et al., 2020, Theorem 4) to sums of randomized functions.

**Proposition 2.** *Let $S = \{Z_1, Z_2, ..., Z_N\}$ be a set of independent random variables valued in $\mathcal{Z}$ and $\xi$ be a random variable valued in $\mathcal{R}$. Let $g_1, ..., g_N$ be a set of measurable functions $g_i : \mathcal{Z}^N \times \mathcal{R} \mapsto \mathbb{R}$ that satisfy the following conditions for any $i \in [N]$:*

- $\mathbb{E}\left[g_i(S, \xi) \mid S \setminus Z_i, \xi\right] = 0$ *and* $|\mathbb{E}[g_i(S, \xi) \mid Z_i, \xi]| \leq M$, *almost surely;*

- $g_i(S, \xi)$ *has the following $L_2$-bounded-difference property with respect to all variables in $S$ except $Z_i$, i.e., $\forall j \neq i$,*

$$\sup_{S, Z_j'} \mathbb{E}_\xi \left[ \left( g_i(S, \xi) - g_i(S^{(j)}, \xi) \right)^2 \right] \leq \beta^2.$$

*Then for all $q \geq 2$,*

$$\left\| \mathbb{E}_\xi \left[ \left| \sum_{i=1}^N g_i(S, \xi) \right| \right] \right\|_q \leq 3M\sqrt{3Nq} + 38N\lceil \log_2 N \rceil \beta q.$$

*Proof in sketch.* The main idea is inspired by the sample-splitting arguments of Feldman and Vondrák (2019); Bousquet et al. (2020), with some new ingredients developed to handle the first-moment operator taken over the internal randomness of functions. Here we just highlight a fundamental difference, which arises from using a newly developed moment inequality (Lemma 9) for bounding the sums of *conditionally independent randomized functions* inside each individual data splits. Different from the version of Marcinkiewicz-Zygmund's inequality used in the original analysis of Bousquet et al. (2020), our new bound in Lemma 9 relies on some second-order (over the function's randomness) components which might be tightly bounded by the second-order concentration inequality in Proposition 1. A full proof is provided in Appendix A.3. □

**Remark 3.** *For sums of deterministic functions, our result in Proposition 2 reduces to the existing moment bound of Bousquet et al. (2020, Theorem 4) which is known to be near-tight up to logarithmic factors. We comment in passing that the tightness analysis of Bousquet et al. (2020, Proposition 9) for deterministic functions can be more or less straightforwardly extended to randomized functions.*

**Remark 4.** *The bound of Proposition 2 would still be valid when the bounded-loss condition $|\mathbb{E}[g_i(S, \xi) \mid Z_i, \xi]| \leq M$ is relaxed to certain sub-Gaussian or sub-exponential stochastic versions.*

## 2.4 Main result on generalization bound

Consequently from Proposition 2, we can now establish our main result on the generalization bound of $L_2$-uniformly stable randomized learning algorithms.

**Theorem 1.** *Let $A : \mathcal{Z}^N \times \mathcal{R} \mapsto \mathcal{W}$ be a randomized learning algorithm that has $L_2$-uniform stability with parameter $\gamma_{L_2, N}$. Assume that the loss function $\ell$ is valued in $[0, M]$. Then for any $\delta \in (0, 1)$, the following bound holds with probability at least $1 - \delta$ over the draw of $S$:*

$$\mathbb{E}_\xi \left[ |R(A(S, \xi)) - R_S(A(S, \xi))| \right] \lesssim \gamma_{L_2, N} \log(N) \log\left( \frac{1}{\delta} \right) + M\sqrt{\frac{\log(1/\delta)}{N}}.$$

*Proof.* See Appendix A.4 for a proof of this result. □

**Remark 5.** *The first-moment bound in Theorem 1 naturally implies the on-average bound in (4) with an identical rate of convergence, though the former is obtained under the relatively stronger notion of $L_2$-uniform stability. As we will see shortly that the $L_2$-uniform stability can indeed be fulfilled by the popularly applied SGD algorithm and thus Theorem 1 is of practical importance for showcasing sharper generalization performance of SGD. When $A$ is deterministic, our bound reduces to the near-optimal (up to logarithmic factors on sample size and failure tail) generalization bound for uniformly stable algorithms (Bousquet et al., 2020).*

---
**Algorithm 1:** Confidence-Boosting for Randomized Learning Algorithms

---
**Input**   : Randomized learning algorithm $A$, data set $S = \{Z_i\}_{i \in [N]}$, $\mu \in (0, 1)$ and $K \in \mathbb{Z}^+$.
**Output** : $A(S, \xi_{k^*})$.
Uniformly divide $S$ into two disjoint subsets $S_1$ and $S_2$ with $|S_1| = (1 - \mu)N, |S_2| = \mu N$.
**for** $k = 1, 2, ..., K$ **do**
| Estimate $A(S_1, \xi_k)$ as an output of $A$ over the subset $S_1$ with random bit $\xi_k$.
**end**
Select the random bit $k^*$ according to $k^* = \arg\min_{k \in [K]} |R_{S_2}(A(S_1, \xi_k)) - R_{S_1}(A(S_1, \xi_k))|$.

---

In view of the standard risk decomposition, the following excess risk tail bound can be readily obtained via applying Theorem 1 and Hoeffding's inequality:

$$\mathbb{E}_\xi \left[ R(A(S, \xi)) - R^* \right] \lesssim \Delta_{\text{opt}} + \gamma_{L_2, N} \log(N) \log\left(\frac{1}{\delta}\right) + M\sqrt{\frac{\log(1/\delta)}{N}}. \tag{13}$$

Here recall that $\Delta_{\text{opt}} := \mathbb{E}_\xi \left[ R_S(A(S, \xi)) \right] - \min_{w \in \mathcal{W}} R_S(w)$ is the sub-optimality of empirical risk minimization. Since the excess risk is by definition non-negative, the above bound can also be obtained under the weaker notion of on-average uniform stability (2) via applying (4). In this sense, the first-moment generalization error bound in Theorem 1 is substantially more challenging to derive than the excess risk bound. Additionally, under the generalized Bernstein condition (Koltchinskii, 2006), the risk bound (13) can be readily improved to (5) by directly applying the corresponding deviation optimal risk bound of Klochkov and Zhivotovskiy (2021) to the on-average loss function $\mathbb{E}_\xi[\ell(A(S, \xi); Z)]$ under on-average uniform stability condition.

## 3   Boosting the Confidence of Generalization

The confidence-boosting technique of Schapire (1990) is a classic meta approach that allows one to boost the dependence of a learning algorithm on the failure probability $\delta$ from $1/\delta$ to $\log(1/\delta)$, at a certain cost of computational complexity. In this section, we show an implication of our first-moment bound in Theorem 1 for achieving high-probability generalization jointly over the randomness of data and algorithm, inside a natural framework of confidence-boosting.

### 3.1   Confidence boosting via bagging

Given a randomized learning algorithm $A$, we propose to study a bagging based confidence-boosting procedure as outlined in Algorithm 1. In this meta procedure, we independently run the algorithm $A$ for $K$ times over $S_1$, a fraction of the training set, to obtain $K$ different candidate solutions $\{A(S_1, \xi_k)\}_{k \in [K]}$. Then we evaluate the validation error of these candidate solutions over the holdout training subset $S_2$, and cherry pick $A(S_1, \xi_{k^*})$ that has the smallest gap between the training error and validation error, i.e., $k^* = \arg\min_{k \in [K]} |R_{S_2}(A(S_1, \xi_k)) - R_{S_1}(A(S_1, \xi_k))|$. Particularly, consider that the internal randomness of $A$ arises from random sampling with replacement of data points, such as SGD under with-replacement sampling. Then in this setting, the procedure can be regarded as a version of bagging (Breiman, 1996) with a greedy model ensemble scheme, which is invoked to the deterministic counterpart of $A$ with fixed random bits (e.g., SGD with identity permutation) over the training subset $S_1$.

### 3.2   Jointly exponential bounds

The following theorem is our main result about the generalization error bound of the output $A(S_1, \xi_{k^*})$ that holds with high probability over the entire training set $S$ and the random seeds $\{\xi_k\}_{k \in [K]}$.

**Theorem 2.** *Suppose that a randomized learning algorithm $A : \mathcal{Z}^N \times \mathcal{R} \mapsto \mathcal{W}$ has $L_2$-uniform stability with parameter $\gamma_{L_2, N}$. Assume that the loss function $\ell$ is valued in $[0, M]$. Then for any $\delta \in (0, 1)$ and $K \geq 2 \log(\frac{2}{\delta})$, with probability at least $1 - \delta$ over the randomness of $S$ and $\{\xi_k\}_{k \in [K]}$, the output of Algorithm 1 satisfies*

$$|R(A(S_1, \xi_{k^*})) - R_S(A(S_1, \xi_{k^*}))| \lesssim \gamma_{L_2, (1-\mu)N} \log(N) \log\left(\frac{1}{\delta}\right) + \frac{M}{\sqrt{\mu(1-\mu)}} \sqrt{\frac{\log(K/\delta)}{N}}.$$

---

**Algorithm 2:** $A_{\texttt{SGD-w}}$: SGD under With-Replacement Sampling

---

**Input** :Data set $S = \{Z_i\}_{i \in [N]}$, step-sizes $\{\eta_t\}_{t \geq 1}$, #iterations $T$, initialization $w_0$.

**Output** :$\bar{w}_T = \frac{1}{T} \sum_{t \in [T]} w_t$.

**for** $t = 1, 2, ..., T$ **do**

  Uniformly randomly sample an index $i_t \in [N]$ with replacement;
  Compute $w_t = \Pi_{\mathcal{W}} (w_{t-1} - \eta_t \nabla_w \ell(w_{t-1}; Z_{i_t}))$.

**end**

---

*Proof in sketch.* Based on Theorem 1, we first prove an intermediate result to show that the minimal generalization error of the $K$ outputs satisfies $\min_{k \in [K]} |R(A(S_1, \xi_k)) - R_{S_1}(A(S_1, \xi_k))| \lesssim$ $\gamma_{\mathtt{L_2},(1-\mu)N} \log(N) \log\left(\frac{1}{\delta}\right) + \frac{M}{\sqrt{\mu(1-\mu)}} \sqrt{\frac{\log(1/\delta)}{N}}$ provided that $K \gtrsim \log(\frac{1}{\delta})$. Next we show that the used greedy model selection strategy guarantees that the selected $A(S, \xi_{k^*})$ mimics the generalization behavior of that best performer among the $K$ candidates, with a slightly expanded $\log(K/\delta)$ factor representing the overhead of simultaneously bounding the generalization performance of $K$ different candidate solutions over the holdout validation set. Finally the desired bound follows from the union probability argument. See Appendix B.1 for its full proof. □

**Remark 6.** *The bound in Theorem 2 holds with high probability jointly over the randomness of sample and algorithm. Different from the bound in (7) that requires high probability uniform stability, Theorem 2 is valid under a substantially milder notion of $L_2$-uniform stability, though at the cost of multiple running of algorithm for confidence boosting. Compared to the bound in (8) that requires certain conditional uniform stability over the random bits of algorithm, our bound has sharper dependence on the uniform stability parameter yet under a weaker notion of stability.*

**Remark 7.** *Regarding the scale of the factor $1/\sqrt{\mu(1-\mu)}$ in the bound of Theorem 2, if setting $\mu = 0.01$ (i.e., 99% of S are used as $S_1$ for training), then the factor is around $10.05$.*

Concerning the excess risk of Algorithm 1, we consider a slightly modified output $A(S_1, \xi_{k^*})$ such that $k^* = \arg\min_{k \in [K]} R_{S_2}(A(S_1, \xi_k))$. Then based on the in-expectation risk bound (13), we can derive the following excess risk bound under the conditions of Theorem 2 using similar arguments:

$$R(A(S_1, \xi_{k^*})) - R^* \lesssim \Delta_{\text{opt}} + \gamma_{\mathtt{L_2},(1-\mu)N} \log(N) \log\left(\frac{1}{\delta}\right) + \frac{M}{\sqrt{\mu(1-\mu)}} \sqrt{\frac{\log(K/\delta)}{N}}. \quad (14)$$

Again, the above risk bound is still valid under the weaker notion of on-average uniform stability (2).

## 4 Implications for SGD

This section is devoted to demonstrating the implications of Theorem 1 and Theorem 2 for the widely used SGD algorithm and its confidence-boosted versions as well. We focus on a variant of SGD under with-replacement sampling as outlined in Algorithm 2, which we call $A_{\texttt{SGD-w}}$. In what follows, we substantialize $\xi = \{i_t\}_{t \in [T]}$ the sample path of $A_{\texttt{SGD-w}}$ over a given data set, and $\{\xi_k\}_{k \in [K]}$ the $K$ independent copies of $\xi$ when implemented with bagging as shown in Algorithm 1. Our results can also be extended to the without-replacement variant of SGD and the corresponding results are provided in Appendix D for the sake of completeness.

### 4.1 Convex optimization with smooth loss

We first present the following lemma that establishes the $L_2$-uniform stability of $A_{\texttt{SGD-w}}$ with convex and smooth loss functions, such as logistic loss. See Appendix C.2 for its proof.

**Lemma 1.** *Suppose that the loss function $\ell(\cdot; \cdot)$ is convex, G-Lipschitz and L-smooth with respect to its first argument. Assume that $\eta_t \leq 2/L$ for all $t \geq 1$. Then $A_{\texttt{SGD-w}}$ has $L_2$-uniform stability with parameter*

$$\gamma_{L_2,N} = 2G^2 \sqrt{10 \left( \frac{1}{N} \sum_{t=1}^{T} \eta_t^2 + \frac{1}{N^2} \left( \sum_{t=1}^{T} \eta_t \right)^2 \right)}.$$

Given Lemma 1, we can apply Theorem 1 and Theorem 2 to immediately obtain the following generalization result for $A_{\text{SGD-w}}$ and its confidence-boosted version with smooth and convex losses.

**Corollary 1.** *Suppose that the loss function $\ell(\cdot;\cdot) \in [0, M]$ is convex, G-Lipschitz and L-smooth with respect to its first argument. Then for any $\delta \in (0, 1)$, it holds with probability at least $1 - \delta$ over the randomness of S that $\mathbb{E}_\xi\left[|R(A_{\text{SGD-w}}(S, \xi)) - R_S(A_{\text{SGD-w}}(S, \xi))|\right] \lesssim$*

$$G^2 \log(N) \log\left(\frac{1}{\delta}\right) \sqrt{\frac{1}{N} \sum_{t=1}^{T} \eta_t^2 + \frac{1}{N^2} \left(\sum_{t=1}^{T} \eta_t\right)^2} + M\sqrt{\frac{\log(1/\delta)}{N}}.$$

*Moreover, consider Algorithm 1 specified to $A_{\text{SGD-w}}$ with learning rate $\eta_t \leq 2/L$ and $K \asymp \log(\frac{1}{\delta})$. Then with probability at least $1 - \delta$ over the randomness of S and $\{\xi_k\}_{k \in [K]}$, it holds that $|R(A_{\text{SGD-w}}(S_1, \xi_{k^*})) - R_S(A_{\text{SGD-w}}(S_1, \xi_{k^*}))| \lesssim$*

$$G^2 \log(N) \log\left(\frac{1}{\delta}\right) \sqrt{\frac{1}{(1-\mu)N} \sum_{t=1}^{T} \eta_t^2 + \frac{1}{(1-\mu)^2 N^2} \left(\sum_{t=1}^{T} \eta_t\right)^2} + \frac{M}{\sqrt{\mu(1-\mu)}}\sqrt{\frac{\log(1/\delta)}{N}}.$$

**Remark 8.** *For the conventional choice of $\eta_t = \frac{2}{L\sqrt{t}}$, the high-probability (w.r.t. data) generalization bounds in Corollary 1 for SGD and its confidence boosted version are roughly of scale $\mathcal{O}\left(\log(N)\log\left(\frac{1}{\delta}\right)\sqrt{\frac{\log(T)}{N}} + \frac{\sqrt{T}}{N}\right)$, which matches the corresponding $\mathcal{O}\left(\frac{\sqrt{T}}{N}\right)$ in-expectation bound of SGD with smooth and convex losses (Hardt et al., 2016).*

Combining with the standard in-expectation optimization error bound of convex SGD (see, e.g., Shamir and Zhang, 2013), we can show the following excess risk bound of (modified) Algorithm 1 as a direct consequence of the generic bound (14) to $A_{\text{SGD-w}}$ with convex and smooth losses:

$$R(A_{\text{SGD-w}}(S_1, \xi_{k^*})) - R^* \lesssim G^2 \log(N) \log\left(\frac{1}{\delta}\right) \sqrt{\frac{1}{(1-\mu)N} \sum_{t=1}^{T} \eta_t^2 + \frac{1}{(1-\mu)^2 N^2} \left(\sum_{t=1}^{T} \eta_t\right)^2}$$
$$+ \frac{M}{\sqrt{\mu(1-\mu)}}\sqrt{\frac{\log(1/\delta)}{N}} + \frac{D^2(w_0, W^*) + G^2 \sum_{t=1}^{T} \eta_t^2}{\sum_{t=1}^{T} \eta_t},$$

where $W^* := \text{Argmin}_{w \in \mathcal{W}} R(w)$ and $D(w, W^*) = \min_{w^* \in W^*} \|w - w^*\|$. With learning rate $\eta_t = \frac{2}{L\sqrt{t}}$, the right hand side of the above roughly scales as $\mathcal{O}\left(\sqrt{\log(N)\log\left(\frac{1}{\delta}\right)\frac{\log(T)}{N}} + \frac{\sqrt{T}}{N} + \frac{\log(T)}{\sqrt{T}}\right)$ which matches the prior high-probability excess risk bounds of SGD with convex losses (Harvey et al., 2019, Remark 3.7).

## 4.2 Convex optimization with non-smooth loss

Now we turn to study the case where the loss is convex but not necessarily smooth, such as the hinge loss and absolute loss. We first establish the following lemma about the $L_2$-uniform stability parameter of $A_{\text{SGD-w}}$ in the considered setting. See Appendix C.3 for its proof.

**Lemma 2.** *Suppose that the loss function $\ell(\cdot;\cdot)$ is convex and G-Lipschitz with respect to its first argument. Then $A_{\text{SGD-w}}$ has $L_2$-uniform stability with parameter*

$$\gamma_{L_2, N} = G^2 \sqrt{40 \sum_{t=1}^{T} \eta_t^2 + \frac{32}{N^2} \left(\sum_{t=1}^{T} \eta_t\right)^2}.$$

With Lemma 2 in place, we can readily apply Theorem 1 and Theorem 2 to establish the following corollary about the generalization bounds of $A_{\text{SGD-w}}$ and its confidence-boosted version with convex and non-smooth loss functions.

**Corollary 2.** *Suppose that the loss function $\ell(\cdot;\cdot) \in [0, M]$ is convex and G-Lipschitz with respect to its first argument. Then for any $\delta \in (0, 1)$, it holds with probability at least $1 - \delta$ over the randomness of S that $\mathbb{E}_\xi\left[|R(A_{\text{SGD-w}}(S, \xi)) - R_S(A_{\text{SGD-w}}(S, \xi))|\right] \lesssim$*

$$G^2 \log(N) \log\left(\frac{1}{\delta}\right) \sqrt{\sum_{t=1}^{T} \eta_t^2 + \frac{1}{N^2} \left(\sum_{t=1}^{T} \eta_t\right)^2} + M\sqrt{\frac{\log(1/\delta)}{N}}.$$

*Moreover, consider Algorithm 1 specified to $A_{\mathtt{SGD-w}}$ with $K \asymp \log(\frac{1}{\delta})$. Then with probability at least $1 - \delta$ over $S$ and $\{\xi_k\}_{k \in [K]}$, it holds that $|R(A_{\mathtt{SGD-w}}(S_1, \xi_{k^*})) - \hat{R}_S(A_{\mathtt{SGD-w}}(S_1, \xi_{k^*}))| \lesssim$*

$$G^2 \log(N) \log\left(\frac{1}{\delta}\right) \sqrt{\sum_{t=1}^{T} \eta_t^2 + \frac{1}{(1-\mu)^2 N^2} \left(\sum_{t=1}^{T} \eta_t\right)^2} + \frac{M}{\sqrt{\mu(1-\mu)}} \sqrt{\frac{\log(1/\delta)}{N}}.$$

**Remark 9.** *For SGD with decaying learning rates $\eta_t = \frac{1}{\sqrt{Nt}}$, Corollary 2 admits high-probability generalization bounds of scale $\mathcal{O}\left(\log(N) \log\left(\frac{1}{\delta}\right) \sqrt{\frac{\log(T)}{N} + \frac{T}{N^3}} + \sqrt{\frac{\log(1/\delta)}{N}}\right)$. With fixed rates $\eta_t \equiv \eta$, Corollary 2 yields deviation bounds of scale $\mathcal{O}\left(\eta \log(N) \log\left(\frac{1}{\delta}\right)(\sqrt{T} + \frac{T}{N}) + \sqrt{\frac{\log(1/\delta)}{N}}\right)$ which matches the near-optimal rate by Bassily et al. (2020, Theorem 3.3).*

### 4.3 Non-convex optimization with smooth loss

We further study the performance of Algorithm 1 for $A_{\mathtt{SGD-w}}$ with smooth but not necessarily convex loss functions, such as normalized sigmoid loss (Mason et al., 1999). The following lemma estimates the $L_2$-uniform stability of $A_{\mathtt{SGD-w}}$ in the considered setting. See Appendix C.4 for its proof.

**Lemma 3.** *Suppose that the loss function $\ell(\cdot; \cdot)$ is $G$-Lipschitz and $L$-smooth with respect to its first argument. Consider $\eta_t \leq 1/L$. Then $A_{\mathtt{SGD-w}}$ has $L_2$-uniform stability with parameter*

$$\gamma_{L_2,N} = 2G^2 \sqrt{\frac{1}{N} \sum_{t=1}^{T} \exp\left(3L \sum_{\tau=t+1}^{T} \eta_\tau\right) u_t},$$

*where*

$$u_t := \eta_t^2 + 2\eta_t \sum_{\tau=1}^{t-1} \exp\left(L \sum_{i=\tau+1}^{t-1} \eta_i\right) \eta_\tau.$$

Based on Lemma 3, we can invoke Theorem 1 and Theorem 2 to show the following generalization result for $A_{\mathtt{SGD-w}}$ and its confidence-boosted version with non-convex and smooth loss functions.

**Corollary 3.** *Suppose that the loss function $\ell(\cdot; \cdot) \in [0, M]$ is $G$-Lipschitz and $L$-smooth with respect to its first argument. Then for any $\delta \in (0, 1)$, it holds with probability at least $1 - \delta$ over the randomness of $S$ that $\mathbb{E}_\xi \left[|R(A_{\mathtt{SGD-w}}(S, \xi)) - \hat{R}_S(A_{\mathtt{SGD-w}}(S, \xi))|\right] \lesssim$*

$$G^2 \log(N) \log\left(\frac{1}{\delta}\right) \sqrt{\frac{1}{N} \sum_{t=1}^{T} \exp\left(L \sum_{\tau=t+1}^{T} \eta_\tau\right) u_t} + M \sqrt{\frac{\log(1/\delta)}{N}},$$

*where $u_t := \eta_t^2 + 2\eta_t \sum_{\tau=1}^{t-1} \exp(L \sum_{i=\tau+1}^{t-1} \eta_i)\eta_\tau$ for all $t \geq 1$. Moreover, consider Algorithm 1 specified to $A_{\mathtt{SGD-w}}$ with $\eta_t \leq \frac{1}{L}$ and $K \asymp \log(\frac{1}{\delta})$. Then with probability at least $1 - \delta$ over $S$ and $\{\xi_k\}_{k \in [K]}$, it holds that $|R(A_{\mathtt{SGD-w}}(S_1, \xi_{k^*})) - \hat{R}_S(A_{\mathtt{SGD-w}}(S_1, \xi_{k^*}))| \lesssim$*

$$G^2 \log(N) \log\left(\frac{1}{\delta}\right) \sqrt{\frac{1}{(1-\mu)N} \sum_{t=1}^{T} \exp\left(L \sum_{\tau=t+1}^{T} \eta_\tau\right) u_t} + \frac{M}{\sqrt{\mu(1-\mu)}} \sqrt{\frac{\log(1/\delta)}{N}}.$$

**Remark 10.** *For the decaying learning rates $\eta_t = \frac{1}{L\nu t}$ with arbitrary $\nu \geq 1$, the generalization bounds in Corollary 3 are of scale $\mathcal{O}\left(\log(N) \log\left(\frac{1}{\delta}\right) \sqrt{\frac{T^{1/\nu} \log(T)}{\nu N}} + \sqrt{\frac{\log(1/\delta)}{N}}\right)$. For the constant learning rates $\eta_t \equiv \frac{1}{LT}$, the bounds are of scale $\mathcal{O}\left(\log(N) \log\left(\frac{1}{\delta}\right) \sqrt{\frac{\log(1/\delta)}{N}}\right)$.*

## 5 Conclusion

In this paper, we have introduced a novel concept of $L_2$-uniform stability for randomized learning algorithms and proved a strong first-moment generalization bound that holds with high probability over training sample. Equipped with this result, we have further developed a bagging based confidence-boosting procedure and shown that it yields near-optimal generalization bounds with high confidence jointly over the randomness of sample and algorithm. The power of our theory has been demonstrated through an application to SGD with time-decaying learning rates, where sharper generalization bounds have been obtained for both convex and non-convex loss functions.

## Acknowledgments and Disclosure of Funding

The authors sincerely thank the anonymous reviewers and area chairs for their insightful comments on this paper. The research was conducted while XTY worked for Nanjing University of Information Science and Technology and both authors worked for Baidu Cognitive Computing Lab. The work of XTY is also funded in part by the National Key Research and Development Program of China under Grant No. 2018AAA0100400, and in part by the Natural Science Foundation of China (NSFC) under Grant No. U21B2049, 61936005.

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
