# A  Proofs for Section 2

In this section, we provide the technical proofs for the main results stated in Section 2.

## A.1  Auxiliary lemmas

Here we collect a set of preliminary lemmas to be used in our analysis. The following lemma is an $L_q$-norm extension of the celebrated Efron-Stein inequality (see, e.g., Boucheron et al., 2005, Theorem 2).

**Lemma 4** (Generalized Efron-Stein inequality). *Let $S = \{Z_1, ..., Z_N\}$ be a set of independent random variables valued in $\mathcal{Z}$ and $h : \mathcal{Z}^N \mapsto \mathbb{R}$ be some measurable function. Then for all $q \geq 2$,*

$$\|h(S) - \mathbb{E}[h(S)]\|_q \leq \sqrt{3q} \sqrt{\left\| \sum_{i=1}^{N} \left( h(S) - h(S^{(i)}) \right)^2 \right\|_{q/2}}.$$

The following result compares the moments and conditional moments of a random function.

**Lemma 5.** *Let $S = \{Z_1, ..., Z_N\}$ be a set of independent random variables valued in some measure space $\mathcal{Z}$ and $h : \mathcal{Z}^N \mapsto \mathbb{R}$ be some measurable function. Then for all $I \subseteq [N]$ and $q \geq 1$, we have*

$$\|h(S_I)\|_q \leq \|h(S)\|_q = \|\|h\|_q(S_I)\|_q.$$

*Proof.* Recall $h(S_I) = \mathbb{E}[h(S) \mid S_I]$. Using Jensen's inequality we can show that

$$\|h(S_I)\|_q = \left( \mathbb{E}\left[ |\mathbb{E}[h(S) \mid S_I]|^q \right] \right)^{1/q} \leq \left( \mathbb{E}\left[ \mathbb{E}[|h(S)|^q \mid S_I]] \right] \right)^{1/q} = \left( \mathbb{E}[|h(S)|^q] \right)^{1/q} = \|h(S)\|_q.$$

By definition we can also express $\|h(S)\|_q = \left( \mathbb{E}\left[ \mathbb{E}[|h(S)|^q \mid S_I]] \right] \right)^{1/q} = \|\|h(S)\|_q(S_I)\|_q$. The proof is completed. $\qquad\square$

We further need to introduce the concept of weakly self-bounding function to be used in the analysis of second-order concentration bounds.

**Definition 2** (Weakly self-bounding function). *A non-negative function $h : \mathcal{Z}^N \mapsto \mathbb{R}^+ \cup \{0\}$ is said to be weakly $(a, b)$-self-bounding with parameters $a, b \geq 0$ if there exist non-negative functions $h_i : \mathcal{Z}^{N-1} \mapsto \mathbb{R}^+ \cup \{0\}$ such that for all $S = \{Z_1, ..., Z_n\} \in \mathcal{Z}^N$,*

$$\sum_{i=1}^{N} \left( h(S) - h_i(S^{\setminus i}) \right)^2 \leq ah(S) + b,$$

*where $S^{\setminus i} := S \setminus \{Z_i\}$.*

The following lemma is a combination of the upper tail bound of Boucheron et al. (2005, Theorem 6.19) and lower tail bound of Klochkov and Zhivotovskiy (2021, Proposition 3.1) for weakly self-bounding functions.

**Lemma 6.** *Let $S = \{Z_1, Z_2, ..., Z_N\}$ be a set of independent random variables valued in $\mathcal{Z}$ and $h : \mathcal{Z}^N \mapsto \mathbb{R}^+ \cup \{0\}$ be a weakly $(a, b)$-self-bounding function.*

- *Assume that the relevant $h_i$ satisfy $h_i(S^{\setminus i}) \leq h(S)$ for any $i = 1, ..., N$. Then for all $t > 0$,*

$$\mathbb{P}\left\{ h(S) \geq \mathbb{E}[h(S)] + t \right\} \leq \exp\left( -\frac{t^2}{2a\mathbb{E}(h(S)) + 2b + at} \right).$$

- *Assume that the relevant $h_i$ satisfy $h_i(S^{\setminus i}) \geq h(S)$ for any $i = 1, ..., N$. Then for all $t > 0$,*

$$\mathbb{P}\left\{ h(S) \leq \mathbb{E}[h(S)] - t \right\} \leq \exp\left( -\frac{t^2}{2a\mathbb{E}(h(S)) + 2b} \right).$$

We also need the following preliminary result about the equivalence between tails and moments (Bousquet et al., 2020).

**Lemma 7.** *Let $Y$ be a real-valued random variable.*

- *If $Y$ satisfies the following inequality for some $a, b \geq 0$ with probability at least $1 - \delta$ for any $\delta \in (0, 1)$,*

$$|Y| \leq a \log\left(\frac{e}{\delta}\right) + b\sqrt{\log\left(\frac{e}{\delta}\right)}.$$

*Then, for any $q \geq 1$ it holds that*

$$\|Y\|_q \leq 3aq + 9b\sqrt{q}.$$

- *If $Y$ satisfies $\|Y\|_q \leq aq + b\sqrt{q}$ for any $q \geq 1$. Then the following holds with probability at least $1 - \delta$ for any $\delta \in (0, 1)$:*

$$|Y| \leq e\left(a \log\left(\frac{e}{\delta}\right) + b\sqrt{\log\left(\frac{e}{\delta}\right)}\right).$$

## A.2 Proof of Proposition 1

The following lemma is key to our proof.

**Lemma 8.** *Let $S = \{Z_1, Z_2, ..., Z_N\}$ be a set of independent random variables valued in $\mathcal{Z}$ and $\xi$ be a random variable valued in $\mathcal{R}$. Let $g : \mathcal{Z}^N \times \mathcal{R} \mapsto \mathbb{R}$ be a measurable function. Then it holds that*

$$\mathbb{E}\left[(g(S, \xi) - \mathbb{E}_S[g(S, \xi)])^2\right] \leq \sum_{i=1}^{N} \mathbb{E}\left[\left(g(S, \xi) - g(S^{(i)}, \xi)\right)^2\right]. \tag{15}$$

*Moreover, for any $q \geq 2$, the following bound holds:*

$$\left\|\mathbb{E}_\xi\left[|g(S, \xi) - \mathbb{E}_S[g(S, \xi)]|\right]\right\|_q$$
$$\leq \sqrt{\sum_{i=1}^{N} \mathbb{E}\left[\left(g(S, \xi) - g(S^{(i)}, \xi)\right)^2\right]} + \sqrt{3q}\sqrt{\left\|\sum_{i=1}^{N} \mathbb{E}_\xi^2\left[|g(S, \xi) - g(S^{(i)}, \xi)|\right]\right\|_{q/2}}. \tag{16}$$

*Proof.* To prove the variance bound (15), we consider the following conditional expectation operators, conditioned on the random variables $(Z_1, ..., Z_i)$ and the random bit $\xi$ of algorithm:

$$f_i := \mathbb{E}\left[g(S, \xi) \mid Z_1, ..., Z_i, \xi\right], \quad \forall i = 1, ..., N.$$

We conventionally define $f_0 = \mathbb{E}_S[g(S, \xi)]$. Clearly, the following telescope decomposition holds:

$$g(S, \xi) - \mathbb{E}_S[g(S, \xi)] = f_N - f_0 = \sum_{i=1}^{N} \{\Delta_i := f_i - f_{i-1}\}.$$

Then we have

$$\mathbb{E}\left[(g(S, \xi) - \mathbb{E}_S[g(S, \xi)])^2\right]$$
$$= \mathbb{E}\left[\left(\sum_{i=1}^{N} \Delta_i\right)^2\right] = \sum_{i=1}^{N} \mathbb{E}[\Delta_i^2] + 2\sum_{i<j} \mathbb{E}[\Delta_i \Delta_j] = \sum_{i=1}^{N} \mathbb{E}[\Delta_i^2], \tag{17}$$

where in the last equality we have used the fact that for any index pair $i < j$, $\mathbb{E}\left[\Delta_j \mid Z_1, ..., Z_i, \xi\right] = 0$ which implies $\mathbb{E}[\Delta_i \Delta_j] = \mathbb{E}\left[\Delta_i \mathbb{E}\left[\Delta_j \mid Z_1, ..., Z_i, \xi\right]\right] = 0$. Note that,

$$\mathbb{E}[\Delta_i^2] = \mathbb{E}\left[\left(\mathbb{E}\left[g(S, \xi) \mid Z_1, ..., Z_i, \xi\right] - \mathbb{E}\left[g(S, \xi) \mid Z_1, ..., Z_{i-1}, \xi\right]\right)^2\right]$$
$$\overset{\varsigma_1}{=} \mathbb{E}\left[\left(\mathbb{E}\left[g(S, \xi) - g(S^{(i)}, \xi)\right]\right)^2 \mid Z_1, ..., Z_i, \xi\right]$$
$$\overset{\varsigma_2}{\leq} \mathbb{E}\left[\mathbb{E}\left[\left(g(S, \xi) - g(S^{(i)}, \xi)\right)^2\right] \mid Z_1, ..., Z_i, \xi\right]$$
$$= \mathbb{E}\left[\left(g(S, \xi) - g(S^{(i)}, \xi)\right)^2\right], \tag{18}$$

where "$\zeta_1$" makes use of the independence of $\xi, Z_1, ..., Z_i, Z'_i$, "$\zeta_2$" is due to Jensen's inequality. Substituting (18) into (17) yields

$$\mathbb{E}\left[(g(S,\xi) - \mathbb{E}_S[g(S,\xi)])^2\right] \leq \sum_{i=1}^{N} \mathbb{E}\left[\left(g(S,\xi) - g(S^{(i)},\xi)\right)^2\right],$$

which is the first desired bound.

We now proceed to prove the $q$-moment bound (16). Let us define $h(S) := \mathbb{E}_\xi\left[|g(S,\xi) - \mathbb{E}_S[g(S,\xi)]|\right]$. Based on Jensen's inequality and triangle inequality we can show that

$$\left|h(S) - h(S^{(i)})\right| = \left|\mathbb{E}_\xi\left[|g(S,\xi) - \mathbb{E}_S[g(S,\xi)]| - \left|g(S^{(i)},\xi) - \mathbb{E}_{S^{(i)}}[g(S^{(i)},\xi)]\right|\right]\right|$$

$$\leq \mathbb{E}_\xi\left[\left|g(S,\xi) - \mathbb{E}_S[g(S,\xi)] - g(S^{(i)},\xi) + \mathbb{E}_{S^{(i)}}[g(S^{(i)},\xi)]\right|\right]$$

$$= \mathbb{E}_\xi\left[\left|g(S,\xi) - g(S^{(i)},\xi)\right|\right].$$

Then by invoking the generalized Efron-Stein inequality (Lemma 4) to $h(S)$ we get that for all $q \geq 2$,

$$\|h(S) - \mathbb{E}_S[h(S)]\|_q \leq \sqrt{3q}\sqrt{\left\|\sum_{i=1}^{N}\left(h(S) - h(S^{(i)})\right)^2\right\|_{q/2}}$$

$$\leq \sqrt{3q}\sqrt{\left\|\sum_{i=1}^{N}\mathbb{E}_\xi^2\left[|g(S,\xi) - g(S^{(i)},\xi)|\right]\right\|_{q/2}}.$$

It follows that

$$\|h(S)\|_q \leq |\mathbb{E}_S[h(S)]| + \sqrt{3q}\sqrt{\left\|\sum_{i=1}^{N}\mathbb{E}_\xi^2\left[|g(S,\xi) - g(S^{(i)},\xi)|\right]\right\|_{q/2}}$$

$$= \mathbb{E}\left[|g(S,\xi) - \mathbb{E}_S[g((S,\xi)]|\right] + \sqrt{3q}\sqrt{\left\|\sum_{i=1}^{N}\mathbb{E}_\xi^2\left[|g(S,\xi) - g(S^{(i)},\xi)|\right]\right\|_{q/2}}$$

$$\leq \sqrt{\sum_{i=1}^{N}\mathbb{E}\left[\left(g(S,\xi) - g(S^{(i)},\xi)\right)^2\right]} + \sqrt{3q}\sqrt{\left\|\sum_{i=1}^{N}\mathbb{E}_\xi^2\left[|g(S,\xi) - g(S^{(i)},\xi)|\right]\right\|_{q/2}},$$

where in the last inequality we have used Jensen's inequality and (15). This gives the desired $q$-moment bound in the second part. $\qquad \square$

**Remark 11.** *The first variance bound in (15) can be regarded as a natural extension of the Efron-Stein inequality to randomized functions.*

Based on Lemma 8, we can prove the main result in Proposition 1.

*Proof of Proposition 1.* The concentration bound (11) can be implied by (16) and the bounded-difference condition as in the following:

$$\|\mathbb{E}_\xi[|g(S,\xi) - \mathbb{E}_S[g(S,\xi)]|]\|_q$$

$$\leq \sqrt{\sum_{i=1}^{N}\mathbb{E}\left[\left(g(S,\xi) - g(S^{(i)},\xi)\right)^2\right]} + \sqrt{3q}\sqrt{\left\|\sum_{i=1}^{N}\mathbb{E}_\xi^2\left[|g(S,\xi) - g(S^{(i)},\xi)|\right]\right\|_{q/2}}$$

$$\leq \beta\sqrt{N} + \sqrt{3q}\sqrt{N\beta^2} \leq 3\beta\sqrt{Nq}.$$

To prove the second-order concentration bound (12), we first show via the inequality (15) in Lemma 8 and the bounded difference condition that

$$\mathbb{E}_\xi\left[(g(S,\xi) - \mathbb{E}_S[g(S,\xi)])^2\right]$$

$$\leq \sum_{i=1}^{N}\mathbb{E}\left[\left(g(S,\xi) - g(S^{(i)},\xi)\right)^2\right] = \sum_{i=1}^{N}\mathbb{E}_S\left[\mathbb{E}_\xi\left[\left(g(S,\xi) - g(S^{(i)},\xi)\right)^2\right]\right] \leq N\beta^2. \tag{19}$$

Let us define $h(S) := \mathbb{E}_\xi\left[(g(S,\xi) - \mathbb{E}_S[g(S,\xi)])^2\right]$. Let $h_i^-(S^{\setminus i}) := \inf_{Z_i \in \mathcal{Z}} h(S)$ such that $h_i^- \le h$. It can be shown that

$$\sum_{i=1}^N \left(h(S) - h_i^-(S^{\setminus i})\right)^2$$
$$= \sum_{i=1}^N \left(\mathbb{E}_\xi\left[(g(S,\xi) - \mathbb{E}_S[g(S,\xi)])^2\right] - \inf_{Z_i \in \mathcal{Z}}\mathbb{E}_\xi\left[(g(S,\xi) - \mathbb{E}_S[g(S,\xi)])^2\right]\right)^2$$
$$\overset{\zeta_1}{\le} N\beta^2\left(\beta + 2\mathbb{E}_\xi\left[|g(S,\xi) - \mathbb{E}_S[g(S,\xi)]|\right]\right)^2$$
$$\le 8N\beta^2 h(S) + 2N\beta^4,$$

where in "$\zeta_1$" we have used Jensen's inequality, Cauchy-Schwarz inequality and the bounded difference assumption. Therefore $h$ is a weakly $(8N\beta^2, 2N\beta^4)$-self-bounding function. Then for any $\delta \in (0,1)$, it follows from the first upper tail bound in Lemma 6 that the following bound holds with probability at least $1 - \frac{\delta}{2}$:

$$h(S) \le \mathbb{E}[h(S)] + 8N\beta^2\log\left(\frac{2}{\delta}\right) + 2\sqrt{(4N\beta^2\mathbb{E}[h(S)] + N\beta^4)\log\left(\frac{2}{\delta}\right)},$$

Now consider $h_i^+(S^{\setminus i}) := \sup_{Z_i \in \mathcal{Z}} h(S)$ such that $h_i^+ \ge h$. Similar to the previous arguments we can show according to the second lower tail bound in Lemma 6 that with probability at least $1 - \frac{\delta}{2}$,

$$h(S) \ge \mathbb{E}[h(S)] - 2\sqrt{(4N\beta^2\mathbb{E}[h(S)] + N\beta^4)\log\left(\frac{2}{\delta}\right)}.$$

Combing the preceding two inequalities yields that the following holds with probability at least $1 - \delta$

$$|h(S) - \mathbb{E}[h(S)]| \le 8N\beta^2\log\left(\frac{2}{\delta}\right) + 2\sqrt{(4N\beta^2\mathbb{E}[h(S)] + N\beta^4)\log\left(\frac{2}{\delta}\right)}.$$

In view of Lemma 7 we further have that for any $q \ge 1$,

$$\|h(S) - \mathbb{E}[h(S)]\|_q \le 24N\beta^2 q + 18\sqrt{(4N\beta^2\mathbb{E}[h(S)] + N\beta^4)\,q}.$$

Consequently we have

$$\|h(S)\|_q \le \mathbb{E}[h(S)] + 24N\beta^2 q + 9\left(\mathbb{E}[h(S)] + \frac{\beta^2}{4} + 4N\beta^2 q\right)$$
$$\le 10\mathbb{E}[h(S)] + 63N\beta^2 q \le 10N\beta^2 + 63N\beta^2 q \le 68N\beta^2 q,$$

where we have used the fact $2ab \le a^2 + b^2$, (19) and $q \ge 2$. The proof is completed. $\qquad\square$

### A.3   Proof of Proposition 2

We need the following lemma as another important and useful consequence of Lemma 8 which can be regarded as an extension of the Marcinkiewicz-Zygmund inequality (Chow and Teicher, 2003) to conditionally independent functions.

**Lemma 9.** *Let $Z_1, Z_2, ..., Z_N$ be a set of independent random variables valued in $\mathcal{Z}$ and $\xi$ be a random variable valued in $\mathcal{R}$. For any $i \in [N]$, let $f_i : \mathcal{Z} \times \mathcal{R} \mapsto \mathbb{R}$ be a measurable function satisfies $\mathbb{E}[f_i(Z_i,\xi) \mid \xi] = 0$. Then for any $q \ge 2$,*

$$\left\|\mathbb{E}_\xi\left[\left|\sum_{i=1}^N f_i(Z_i,\xi)\right|\right]\right\|_q \le \sqrt{2\sum_{i=1}^N\mathbb{E}\left[f_i^2(Z_i,\xi)\right]} + 2\sqrt{3q}\sqrt{\left\|\sum_{i=1}^N\mathbb{E}_\xi^2\left[|f_i(Z_i,\xi)|\right]\right\|_{q/2}}. \qquad (20)$$

*Proof.* Let $S = \{Z_1, Z_2, ..., Z_N\}$ and consider $g(S, \xi) = \sum_{i=1}^{N} f_i(Z_i, \xi)$. Then it can be verified that $\mathbb{E}_S [g(S, \xi)] = 0$,

$$\mathbb{E}\left[\left(g(S, \xi) - g(S^{(i)}, \xi)\right)^2\right] = \mathbb{E}\left[(f_i(Z_i, \xi) - f_i(Z_i', \xi))^2\right] = 2\mathbb{E}\left[f_i^2(Z_i, \xi)\right],$$

and

$$\mathbb{E}_\xi^2\left[\left|g(S, \xi) - g(S^{(i)}, \xi)\right|\right] = \mathbb{E}_\xi^2\left[|f_i(Z_i, \xi) - f_i(Z_i', \xi)|\right] \leq 2\mathbb{E}_\xi^2\left[|f_i(Z_i, \xi)|\right] + 2\mathbb{E}_\xi^2\left[|f_i(Z_i', \xi)|\right].$$

Applying Lemma 8 to $g(S, \xi)$ yields

$$\left\|\mathbb{E}_\xi\left[\left|\sum_{i=1}^{N} f_i(Z_i, \xi)\right|\right]\right\|_q = \|\mathbb{E}_\xi\left[|g(S, \xi) - \mathbb{E}_S[g(S, \xi)]|\right]\|_q$$

$$\leq \sqrt{2\sum_{i=1}^{N} \mathbb{E}\left[f_i^2(Z_i, \xi)\right]} + \sqrt{3q}\sqrt{\left\|2\sum_{i=1}^{N}\left(\mathbb{E}_\xi^2\left[|f_i(Z_i, \xi)|\right] + \mathbb{E}_\xi^2\left[|f_i(Z_i', \xi)|\right]\right)\right\|_{q/2}}$$

$$= \sqrt{2\sum_{i=1}^{N} \mathbb{E}\left[f_i^2(Z_i, \xi)\right]} + 2\sqrt{3q}\sqrt{\left\|\sum_{i=1}^{N} \mathbb{E}_\xi^2\left[|f_i(Z_i, \xi)|\right]\right\|_{q/2}}.$$

This proves the desired bound. $\square$

We also need the following lemma which indicates that conditional expectation does not expand the $L_2$-bounded-difference parameter.

**Lemma 10.** *Let $S = \{Z_1, Z_2, ..., Z_N\}$ be a set of independent random variables valued in $\mathcal{Z}$ and $\xi$ be a random variable valued in $\mathcal{R}$. Let $g : \mathcal{Z}^N \times \mathcal{R} \mapsto \mathbb{R}$ be some measurable function. Let $I \subseteq [N]$ be an index set. Then for all $i \in I$,*

$$\sup_{S_I, S_I^{(i)}} \sqrt{\mathbb{E}_\xi\left[\left(g(S_I, \xi) - g(S_I^{(i)}, \xi)\right)^2\right]} \leq \sup_{S, S^{(i)}} \sqrt{\mathbb{E}_\xi\left[\left(g(S, \xi) - g(S^{(i)}, \xi)\right)^2\right]}$$

*Proof.* Recall that $g(S_I, \xi) = \mathbb{E}[g(S, \xi) \mid S_I, \xi]$. Based on Jensen's inequality we can show that

$$\mathbb{E}_\xi\left[\left(g(S_I, \xi) - g(S_I^{(i)}, \xi)\right)^2\right] \leq \mathbb{E}_\xi\left[\mathbb{E}\left[\left(g(S, \xi) - g(S^{(i)}, \xi)\right)^2 \mid S_I, S_I^{(i)}, \xi\right]\right]$$

$$= \mathbb{E}\left[\mathbb{E}_\xi\left[\left(g(S, \xi) - g(S^{(i)}, \xi)\right)^2\right] \mid S_I, S_I^{(i)}\right]$$

$$\leq \sup_{S, S^{(i)}} \mathbb{E}_\xi\left[\left(g(S, \xi) - g(S^{(i)}, \xi)\right)^2\right],$$

where in the last inequality we have used the fact that expectation is always no larger than maximum. The desired bound then follows immediately from the above inequality. $\square$

Now we are in the position to prove the main result. The proof is inspired by the sample-splitting arguments of Feldman and Vondrák (2019); Bousquet et al. (2020), with several non-trivial modifications along made to deal with the challenges arisen from the first-moment operation taken over the randomness of algorithm.

*Proof of Proposition 2.* Consider $k$ such that $2^{k-1} < N \leq 2^k$. If $N < 2^k$, we pad the training set $S$ with extra zero-functions so that $N = 2^k$. Consider the partition $\mathcal{I}_0, \mathcal{I}_1, ..., \mathcal{I}_k$ of $[N]$ given by

$$\mathcal{I}_0 = \{\{1\}, ..., \{2^k\}\}, \quad \mathcal{I}_1 = \{\{1, 2\}, \{3, 4\}..., \{2^k - 1, 2^k\}\}, \quad \mathcal{I}_k = \{\{1, ..., 2^k\}\}.$$

For any $i \in [N]$ and $l = 0, ..., k$, we denote by $I^l(i) \in \mathcal{I}_l$ the only set from $\mathcal{I}_l$ that contains $i$ and consider the following random variables:

$$g_i^l = \mathbb{E}\left[g_i \mid Z_i, S_{\overline{I^l(i)}}, \xi\right].$$

In particular, $g_i^0 = g_i$ and $g_i^k = \mathbb{E}[g_i \mid Z_i, \xi]$. Clearly we have the following telescope sum:

$$g_i = \sum_{l=0}^{k-1}(g_i^l - g_i^{l+1}) + g_i^k = \sum_{l=0}^{k-1}(g_i^l - g_i^{l+1}) + \mathbb{E}[g_i \mid Z_i, \xi].$$

It follows that

$$\left\| \mathbb{E}_\xi \left[ \left| \sum_{i=1}^N (g_i - \mathbb{E}[g_i \mid Z_i, \xi]) \right| \right] \right\|_q \leq \sum_{l=0}^{k-1} \left\| \mathbb{E}_\xi \left[ \left| \sum_{i=1}^N g_i^l - g_i^{l+1} \right| \right] \right\|_q. \tag{21}$$

We need to upper bound the r.h.s. of the above inequality. To this end, it can be verified that

$$g_i^{l+1} = \mathbb{E}\left[ g_i \mid Z_i, S_{\overline{I^{l+1}(i)}}, \xi \right] = \mathbb{E}\left[ g_i^l \mid Z_i, S_{\overline{I^{l+1}(i)}}, \xi \right].$$

Since by assumption $g_i$ has bounded $L_2$-uniform difference by $\beta$ with respect to all variables except the $i$-th variable, it can be verified by Lemma 10 that so is $g_i^l$ for each $l = 0, ..., k$. Conditioned on $Z_i, S_{\overline{I^{l+1}(i)}}$, invoking Theorem 1 to $g_i^l$ yields that for any $q \geq 1$,

$$\left\| \mathbb{E}_\xi \left[ |g_i^l - g_i^{l+1}| \right] \right\|_q \left( Z_i, S_{\overline{I^{l+1}(i)}} \right) \leq 3\beta\sqrt{q2^l},$$
$$\left\| \mathbb{E}_\xi \left[ (g_i^l - g_i^{l+1})^2 \right] \right\|_q \left( Z_i, S_{\overline{I^{l+1}(i)}} \right) \leq 68\beta^2 2^l q,$$

as there are $2^l$ indices in $I^{l+1}(i) \setminus I^l(i)$. Consequently according to Lemma 5 we have that for any $q \geq 1$,

$$\left\| \mathbb{E}_\xi \left[ |g_i^l - g_i^{l+1}| \right] \right\|_q = \left\| \left\| \mathbb{E}_\xi |g_i^l - g_i^{l+1}| \right\|_q \left( Z_i, S_{\overline{I^{l+1}(i)}} \right) \right\|_q \leq 3\beta\sqrt{q2^l}, \tag{22}$$

and

$$\left\| \mathbb{E}_\xi \left[ (g_i^l - g_i^{l+1})^2 \right] \right\|_q = \left\| \left\| \mathbb{E}_\xi \left[ (g_i^l - g_i^{l+1})^2 \right] \right\|_q \left( Z_i, S_{\overline{I^{l+1}(i)}} \right) \right\|_q \leq 68\beta^2 2^l q. \tag{23}$$

Now consider any $I^l \in \mathcal{I}_l$. Since for each $i \in I_l$, $g_i^l - g_i^{l+1}$ depends only on $(Z_i, S_{\overline{I^l}}, \xi)$, these random terms are essentially of the form $f(Z_i, \xi)$ conditioned on $(S_{\overline{I^l}})$. Given the assumption $\mathbb{E}[g_i(S, \xi) \mid S \setminus Z_i, \xi] = 0$, it holds that $\mathbb{E}[g_i^l - g_i^{l+1} \mid S_{\overline{I^l}}, \xi] = 0, \forall i \in I_l$. Therefore, applying Lemma 9 yields

$$\left\| \mathbb{E}_\xi \left[ \left| \sum_{i \in I^l} g_i^l - g_i^{l+1} \right| \right] \right\|_q (S_{\overline{I^l}})$$
$$\leq \sqrt{2 \sum_{i=1}^N \mathbb{E}\left[ (g_i^l - g_i^{l+1})^2 \mid S_{\overline{I^l}} \right] + 2\sqrt{3q} \sqrt{\left\| \sum_{i=1}^N \mathbb{E}_\xi^2 \left[ |g_i^l - g_i^{l+1}| \right] \right\|_{q/2} (S_{\overline{I^l}})}}.$$

Next we proceed to bound the moment $\left\|\mathbb{E}_\xi\left[\left|\sum_{i\in I^l}g_i^l - g_i^{l+1}\right|\right]\right\|_q$. By applying Lemma 5 again and using the above bound we can show that

$$\left\|\mathbb{E}_\xi\left[\left|\sum_{i\in I^l}g_i^l - g_i^{l+1}\right|\right]\right\|_q = \left\|\left\|\mathbb{E}_\xi\left[\left|\sum_{i\in I^l}g_i^l - g_i^{l+1}\right|\right]\right\|\left(S_{\overline{I^l}}\right)\right\|_q$$

$$\leq \left\|\sqrt{2\sum_{i\in I^l}\mathbb{E}\left[\left(g_i^l - g_i^{l+1}\right)^2 \mid S_{\overline{I^l}}\right] + 2\sqrt{3q}\sqrt{\left\|\sum_{i\in I^l}\mathbb{E}_\xi^2\left[|g_i^l - g_i^{l+1}|\right]\right\|_{q/2}}\left(S_{\overline{I^l}}\right)}\right\|_q$$

$$\leq \left\|\sqrt{2\sum_{i\in I^l}\mathbb{E}\left[\left(g_i^l - g_i^{l+1}\right)^2 \mid S_{\overline{I^l}}\right]}\right\|_q + 2\sqrt{3q}\left\|\sqrt{\left\|\sum_{i\in I^l}\mathbb{E}_\xi^2\left[|g_i^l - g_i^{l+1}|\right]\right\|_{q/2}}\left(S_{\overline{I^l}}\right)\right\|_q$$

$$\overset{\zeta_1}{=}\sqrt{2\left\|\sum_{i\in I^l}\mathbb{E}\left[\left(g_i^l - g_i^{l+1}\right)^2 \mid S_{\overline{I^l}}\right]\right\|_{q/2}} + 2\sqrt{3q}\sqrt{\left\|\left\|\sum_{i\in I^l}\mathbb{E}_\xi^2\left[|g_i^l - g_i^{l+1}|\right]\right\|_{q/2}\left(S_{\overline{I^l}}\right)\right\|_{q/2}}$$

$$\leq \sqrt{2\sum_{i\in I^l}\left\|\mathbb{E}\left[\left(g_i^l - g_i^{l+1}\right)^2 \mid S_{\overline{I^l}}\right]\right\|_{q/2}} + 2\sqrt{3q}\sqrt{\sum_{i\in I^l}\left\|\mathbb{E}_\xi^2\left[|g_i^l - g_i^{l+1}|\right]\right\|_{q/2}}$$

$$\overset{\zeta_2}{\leq}\sqrt{2\sum_{i\in I^l}\left\|\mathbb{E}_\xi\left[\left(g_i^l - g_i^{l+1}\right)^2\right]\right\|_{q/2}} + 2\sqrt{3q}\sqrt{\sum_{i\in I^l}\left\|\mathbb{E}_\xi\left[|g_i^l - g_i^{l+1}|\right]\right\|_q^2}$$

$$\overset{\zeta_3}{\leq}\sqrt{2\times 2^l\times 34\beta^2 2^l q} + 2\sqrt{3q}\sqrt{2^l\times 9\beta^2 2^l q} = 2^l\beta\left(2\sqrt{17q} + 6\sqrt{3}q\right) \leq 2^l\times 19\beta q,$$

where "$\zeta_1$" is due to (10), in "$\zeta_2$" we have used Lemma 5, and in "$\zeta_3$" we have used (22) and (23). Then based on the triangle inequality we get

$$\left\|\mathbb{E}_\xi\left[\left|\sum_{i=1}^N g_i^l - g_i^{l+1}\right|\right]\right\|_q \leq \sum_{I^l\in\mathcal{I}_l}\left\|\mathbb{E}_\xi\left[\left|\sum_{i\in I^l}g_i^l - g_i^{l+1}\right|\right]\right\|_q \leq 2^{k-l}\times 2^l\times 19\beta q \leq 38N\beta q.$$

Therefore the r.h.s. of (21) can be bounded as

$$\left\|\mathbb{E}_\xi\left[\left|\sum_{i=1}^N (g_i - \mathbb{E}[g_i \mid Z_i, \xi])\right|\right]\right\|_q \leq \sum_{l=0}^{k-1}\left\|\mathbb{E}_\xi\left[\left|\sum_{i=1}^N g_i^l - g_i^{l+1}\right|\right]\right\|_q \leq 38N\lceil\log_2 N\rceil\beta q. \quad (24)$$

Based on (24) and the triangle inequality we have

$$\left\|\mathbb{E}_\xi\left[\left|\sum_{i=1}^N g_i\right|\right]\right\|_q \leq \left\|\mathbb{E}_\xi\left[\left|\sum_{i=1}^N \mathbb{E}[g_i \mid Z_i, \xi]\right|\right]\right\|_q + 38N\lceil\log_2 N\rceil\beta q. \quad (25)$$

Let $f_i(Z_i, \xi) = \mathbb{E}[g_i(S, \xi) \mid Z_i, \xi]$. We must have $|f_i| \leq M$ and $\mathbb{E}[f_i(Z_i, \xi) \mid \xi] = 0$ as $|\mathbb{E}[g_i \mid Z_i, \xi]| \leq M$ and $\mathbb{E}[g_i \mid S \setminus Z_i, \xi] = 0$. Then it follows from Lemma 9 that the first term at the right hand side of (25) can be bounded as

$$\left\|\mathbb{E}_\xi\left[\left|\sum_{i=1}^N \mathbb{E}[g_i \mid Z_i, \xi]\right|\right]\right\|_q$$
$$= \left\|\mathbb{E}_\xi\left[\left|\sum_{i=1}^N f_i(Z_i, \xi)\right|\right]\right\|_q \leq M\sqrt{2N} + 2M\sqrt{3Nq} \leq 3M\sqrt{3Nq}. \quad (26)$$

Finally, the desired bound is obtained by plugging (26) into (25). $\qquad\square$

## A.4 Proof of Theorem 1

*Proof.* Let us consider $g_i(S, \xi) = \mathbb{E}_{Z_i'} \left[ R(A(S^{(i)}, \xi)) - \ell(A(S^{(i)}, \xi); Z_i) \right]$. Then the $L_q$-norm of the on-average generalization gap can be upper bounded by the triangle inequality as

$$
\| \mathbb{E}_\xi \left[ | R(A(S, \xi)) - R_S(A(S, \xi)) | \right] \|_q = \frac{1}{N} \left\| \mathbb{E}_\xi \left[ \left| \sum_{i=1}^N R(A(S, \xi)) - \ell(A(S, \xi); Z_i) \right| \right] \right\|_q
$$

$$
\leq \frac{1}{N} \left( \underbrace{\left\| \mathbb{E} \left[ \left| \sum_{i=1}^N g_i(S, \xi) \right| \right] \right\|_q}_{T_1} + \underbrace{\left\| \mathbb{E} \left[ \left| \sum_{i=1}^N (R(A(S, \xi)) - \ell(A(S, \xi); Z_i) - g_i(S, \xi)) \right| \right] \right\|_q}_{T_2} \right). \quad (27)
$$

We next respectively upper bound the two terms $T_1$ and $T_2$ in (27). To bound the term $T_1$, by definition it holds that $\mathbb{E}[g_i(S, \xi) \mid S \setminus Z_i, \xi] = 0$. Based on the triangle inequality and Jensen's inequality we have that for any $i \in [N]$,

$$
|\mathbb{E}[g_i(S, \xi) \mid Z_i, \xi]| \leq \mathbb{E}[\ell(A(S^{(i)}, \xi); Z) \mid \xi] + \mathbb{E}[\ell(A(S^{(i)}, \xi); Z_i) \mid Z_i, \xi] \leq 2M.
$$

Further we show that $g_i$ satisfies the $L_2$-uniform bounded difference property with respect to all variables in $S$ except $Z_i$. Indeed, for each $j \neq i$ and conditioned on $S, Z_j'$ it can be verified that

$$
\left\| g_i(S, \xi) - g_i(S^{(j)}, \xi) \right\|_2 (S, Z_j')
$$

$$
\leq \left\| \mathbb{E}_{Z_i'} \left[ R(A(S^{(i)}, \xi)) - R(A((S^{(i)})^{(j)}, \xi)) \right] \right\|_2 (S, Z_j')
$$

$$
+ \left\| \mathbb{E}_{Z_i'} \left[ \ell(A(S^{(i)}, \xi); Z_i) - \ell(A((S^{(i)})^{(j)}, \xi); Z_i) \right] \right\|_2 (S, Z_j')
$$

$$
= \left\| \mathbb{E}_{Z_i'} \mathbb{E}_Z \left[ \ell(A(S^{(i)}, \xi); Z) - \ell(A((S^{(i)})^{(j)}, \xi); Z) \right] \right\|_2 (S, Z_j')
$$

$$
+ \left\| \mathbb{E}_{Z_i'} \left[ \ell(A(S^{(i)}, \xi); Z_i) - \ell(A((S^{(i)})^{(j)}, \xi); Z_i) \right] \right\|_2 (S, Z_j')
$$

$$
\leq \sup_{Z_i', Z} \left\| \ell(A(S^{(i)}, \xi); Z) - \ell(A((S^{(i)})^{(j)}, \xi); Z) \right\|_2 (S, Z_i', Z_j', Z)
$$

$$
+ \sup_{Z_i'} \left\| \ell(A(S^{(i)}, \xi); Z_i) - \ell(A((S^{(i)})^{(j)}, \xi); Z_i) \right\|_2 (S, Z_i', Z_j')
$$

$$
\leq \sup_{S^{(i)}, Z_j', Z} \left\| \ell(A(S^{(i)}, \xi); Z) - \ell(A((S^{(i)})^{(j)}, \xi); Z) \right\|_2 (S^{(i)}, Z_j', Z)
$$

$$
+ \sup_{S^{(i)}, Z_j', Z_i} \left\| \ell(A(S^{(i)}, \xi); Z_i) - \ell(A((S^{(i)})^{(j)}, \xi); Z_i) \right\|_2 (S^{(i)}, Z_j', Z_i)
$$

$$
\leq 2\gamma_{\mathrm{L_2}, N},
$$

where we have frequently used the fact that expectation is always no larger than supreme, and in the last equality we have used the $L_2$-uniform stability assumption on the algorithm $A$. Therefore, $\{g_i\}$ satisfy the conditions of Proposition 2 and thus

$$
T_1 = \left\| \mathbb{E}_\xi \left[ \left| \sum_{i=1}^N g_i(S, \xi) \right| \right] \right\|_q \leq 6M\sqrt{3Nq} + 76N \lceil \log_2 N \rceil \gamma_{\mathrm{L_2}, N} q. \quad (28)
$$

Now we proceed to bound the second term $T_2$. It can be verified that

$$
T_2 \leq \left\| \mathbb{E}_\xi \left[ \left| \sum_{i=1}^N \mathbb{E}_{Z_i'} \left[ R(A(S,\xi)) - R(A(S^{(i)},\xi)) \right] \right| \right] \right\|_q
$$

$$
+ \left\| \mathbb{E}_\xi \left[ \left| \sum_{i=1}^N \mathbb{E}_{Z_i'} \left[ \ell(A(S,\xi); Z_i) - \ell(A(S^{(i)},\xi); Z_i) \right] \right| \right] \right\|_q
$$

$$
= \left\| \mathbb{E}_\xi \left[ \left| \sum_{i=1}^N \mathbb{E}_{Z_i'} \mathbb{E}_Z \left[ \ell(A(S,\xi); Z) - \ell(A(S^{(i)},\xi); Z) \right] \right| \right] \right\|_q \tag{29}
$$

$$
+ \left\| \mathbb{E}_\xi \left[ \left| \sum_{i=1}^N \mathbb{E}_{Z_i'} \left[ \ell(A(S,\xi); Z_i) - \ell(A(S^{(i)},\xi); Z_i) \right] \right| \right] \right\|_q
$$

$$
\leq 2 \sum_{i=1}^N \left( \sup_{S, Z_i', Z} \mathbb{E}_\xi \left[ \left| \ell(A(S,\xi); Z) - \ell(A(S^{(i)},\xi); Z) \right| \right] \right)
$$

$$
\leq 2N \gamma_{\mathrm{L}_2, N},
$$

where in the last equality we have used the $L_2$-uniform stability assumption. Plugging bounds (28) and (29) into (27) and preserving leading terms yields

$$
\| \mathbb{E}_\xi \left[ |R(A(S,\xi)) - R_S(A(S,\xi))| \right] \|_q \leq 6M \sqrt{\frac{3q}{N}} + 77 \lceil \log_2 N \rceil \gamma_{\mathrm{L}_2, N} q.
$$

According to the equivalence of tails and moments as shown in Lemma 7, the above moment bound immediately implies the desired exponential generalization bound. $\qquad\square$

# B    Proofs for Section 3

In this section, we present the technical proofs for the main results stated in Section 3.

## B.1    Proof of Theorem 2

We first establish the following intermediate result that captures the effects of bagging on randomized algorithms: it basically tells that with $K \gtrsim \log(\frac{1}{\delta})$, at least one of the solutions generated by bagging generalizes well with high probability.

**Lemma 11.** *Suppose that a randomized learning algorithm $A : \mathcal{Z}^N \times \mathcal{R} \mapsto \mathcal{W}$ has $L_2$-uniform stability with parameter $\gamma_{\mathrm{L}_2, N}$. Assume that the loss function $\ell$ is ranged in $[0, M]$. Then for any $\delta \in (0, 1)$ and $K \geq 2\log(2/\delta)$, with probability at least $1 - \delta$ over the randomness of $S_1$ and $\{\xi_k\}_{k \in [K]}$, the sequence $\{A(S_1, \xi_k)\}_{k \in [K]}$ generated by Algorithm 1 satisfies*

$$
\min_{k \in [K]} |R(A(S_1, \xi_k)) - R_{S_1}(A(S_1, \xi_k))| \lesssim \gamma_{\mathrm{L}_2, (1-\mu)N} \log(N) \log\left(\frac{1}{\delta}\right) + M \sqrt{\frac{\log(1/\delta)}{(1-\mu)N}}.
$$

*Proof.* For any data set $S$, let us define $h(S) := \mathbb{E}_\xi \left[ |R(A(S,\xi)) - R_S(A(S,\xi))| \right]$. From Theorem 1 we have that with probability at least $1 - \delta$ over $S_1$,

$$
h(S_1) \lesssim \gamma_{\mathrm{L}_2, (1-\mu)N} \log((1-\mu)N) \log\left(\frac{1}{\delta}\right) + M \sqrt{\frac{\log(1/\delta)}{(1-\mu)N}}. \tag{30}
$$

Let us now consider the following defined events:

$$
\mathcal{E} : \min_{k \in [K]} |R(A(S_1, \xi_k)) - R_{S_1}(A(S_1, \xi_k))| \lesssim \gamma_{\mathrm{L}_2, (1-\mu)N} \log((1-\mu)N) \log\left(\frac{1}{\delta}\right) + M \sqrt{\frac{\log(1/\delta)}{(1-\mu)N}},
$$

$$
\mathcal{E}_0 : h(S_1) \lesssim \gamma_{\mathrm{L}_2, (1-\mu)N} \log(N) \log\left(\frac{1}{\delta}\right) + M \sqrt{\frac{\log(1/\delta)}{(1-\mu)N}},
$$

$$
\mathcal{E}_k : |R(A(S_1, \xi_k)) - R_{S_1}(A(S_1, \xi_k))| \leq 2h(S_1), \quad k = 1, ..., K.
$$

We first show that $\mathbb{P}\left\{\bigcap_{k\in[K]}\overline{\mathcal{E}_k}\right\}\leq\frac{\delta}{2}$. To this end, for each $k$, let us consider the random indication function $g(S_1,\xi_k):=\mathbf{1}_{\{\overline{\mathcal{E}_k}\}}$ where $\mathbf{1}_{\{C\}}$ is the indicator function of the condition $C$. Then we can show that

$$
\begin{aligned}
\mathbb{P}\left\{\bigcap_{k=1}^{K}\overline{\mathcal{E}_k}\right\} &=\mathbb{E}\left[\prod_{k=1}^{K}g(S_1,\xi_k)\right]\\
&=\mathbb{E}\left[\mathbb{E}\left[\prod_{k=1}^{K}g(S_1,\xi_k)\mid S_1\right]\right]\\
&=\mathbb{E}\left[\prod_{k=1}^{K}\mathbb{E}\left[g(S_1,\xi_k)\mid S_1\right]\right]\\
&=\mathbb{E}\left[\prod_{k=1}^{K}\mathbb{P}\left\{|R(A(S_1,\xi_k))-R_{S_1}(A(S_1,\xi_k))|>2h(S_1)\right\}\mid S_1\right]\\
&\leq\mathbb{E}\left[\left(\frac{1}{2}\right)^{K}\mid S_1\right]=\left(\frac{1}{2}\right)^{K}\leq\frac{\delta}{2},
\end{aligned}
$$

where we have used the independence among $\{\xi_k\}$ and $S_1$, Markov inequality and the condition on $K$ as well. From the high-probability bound (30) we have $\mathbb{P}\left\{\overline{\mathcal{E}_0}\right\}\leq\frac{\delta}{2}$. Combining this bound and the preceding bound yields

$$
\mathbb{P}\left\{\mathcal{E}\right\}\geq\mathbb{P}\left\{\left(\bigcup_{k=1}^{K}\mathcal{E}_k\right)\bigcap\mathcal{E}_0\right\}\geq 1-\mathbb{P}\left\{\bigcap_{k=1}^{K}\overline{\mathcal{E}_k}\right\}-\mathbb{P}\left\{\overline{\mathcal{E}_0}\right\}\geq 1-\frac{\delta}{2}-\frac{\delta}{2}=1-\delta.
$$

This proves the desired bound. $\qquad\square$

With Lemma 11 in place, we are ready to prove the main result in Theorem 2.

*Proof of Theorem 2.* The key idea is to show that the proposed greedy model selection strategy guarantees that the selected $A(S,\xi_{k^*})$ mimics the generalization behavior of the best performer among the $K$ candidates. To do so, let us consider the following three events:

$$
\mathcal{E}:\ |R(A(S_1,\xi_{k^*}))-R_S(A(S_1,\xi_{k^*}))|\lesssim\gamma_{\mathrm{L}_2,(1-\mu)N}\log(N)\log\left(\frac{1}{\delta}\right)+\frac{M}{\sqrt{\mu(1-\mu)}}\sqrt{\frac{\log(K/\delta)}{N}},
$$

$$
\mathcal{E}_1:\ \max_{k\in[K]}|R(A(S_1,\xi_k))-R_{S_2}(A(S_1),\xi_k)|\leq M\sqrt{\frac{\log(2K/\delta)}{2\mu N}},
$$

$$
\mathcal{E}_2:\ \min_{k\in[K]}|R(A(S_1,\xi_k))-R_{S_1}(A(S_1,\xi_k))|\lesssim\gamma_{\mathrm{L}_2,(1-\mu)N}\log(N)\log\left(\frac{1}{\delta}\right)+M\sqrt{\frac{\log(1/\delta)}{(1-\mu)N}}.
$$

We first show that $\mathcal{E} \supseteq \mathcal{E}_1 \cap \mathcal{E}_2$. Indeed, suppose that $\mathcal{E}_1$ and $\mathcal{E}_2$ simultaneously occur. Consequently the following inequality can be verified:

$$
\begin{aligned}
&|R(A(S_1, \xi_{k^*})) - R_S(A(S_1, \xi_{k^*}))| \\
={}&|R(A(S_1, \xi_{k^*})) - (1-\mu)R_{S_1}(A(S_1, \xi_{k^*})) - \mu R_{S_2}(A(S_1, \xi_{k^*}))| \\
\leq{}&(1-\mu)|R(A(S_1, \xi_{k^*})) - R_{S_1}(A(S_1, \xi_{k^*}))| + \mu|R(A(S_1, \xi_{k^*})) - R_{S_2}(A(S_1, \xi_{k^*}))| \\
\leq{}&(1-\mu)|R_{S_2}(A(S_1, \xi_{k^*})) - R_{S_1}(A(S_1, \xi_{k^*}))| + |R(A(S_1, \xi_{k^*})) - R_{S_2}(A(S_1, \xi_{k^*}))| \\
\overset{\zeta_1}{=}{}&(1-\mu)\min_{k\in[K]}|R_{S_2}(A(S_1, \xi_k)) - R_{S_1}(A(S_1, \xi_k))| + |R(A(S_1, \xi_{k^*})) - R_{S_2}(A(S_1, \xi_{k^*}))| \\
={}&(1-\mu)\min_{k\in[K]}|R_{S_2}(A(S_1, \xi_k)) - R(A(S_1, \xi_k)) + R(A(S_1, \xi_k)) - R_{S_1}(A(S_1, \xi_k))| \\
&+ |R(A(S_1, \xi_{k^*})) - R_{S_2}(A(S_1, \xi_{k^*}))| \\
\leq{}&(1-\mu)\min_{k\in[K]}|R(A(S_1, \xi_k)) - R_{S_1}(A(S_1, \xi_k))| + (1-\mu)\max_{k\in[K]}|R_{S_2}(A(S_1, \xi_k)) - R(A(S_1, \xi_k))| \\
&+ |R(A(S_1, \xi_{k^*})) - R_{S_2}(A(S_1, \xi_{k^*}))| \\
\leq{}&\min_{k\in[K]}|R(A(S_1, \xi_k)) - R_{S_1}(A(S_1, \xi_k))| + 2\max_{k\in[K]}|R_{S_2}(A(S_1, \xi_k)) - R(A(S_1, \xi_k))| \\
\overset{\zeta_2}{\lesssim}{}&\gamma_{\mathrm{L}_2, (1-\mu)N}\log(N)\log\left(\frac{1}{\delta}\right) + M\sqrt{\frac{\log(1/\delta)}{(1-\mu)N}} + M\sqrt{\frac{\log(K/\delta)}{\mu N}},
\end{aligned}
$$

where in "$\zeta_1$" we have used the definition of $k^*$, and "$\zeta_2$" follows from $\mathcal{E}_1, \mathcal{E}_2$. After some algebraic manipulation with leading terms preserved in the above we can see that $\mathcal{E}$ occurs.

Next we aim to show that $\mathbb{P}\{\overline{\mathcal{E}_1}\} \leq \frac{\delta}{2}$. To this end, let us consider the random indication function $g(S, \{\xi_k\}) := \mathbf{1}_{\{\overline{\mathcal{E}_1}\}}$ associated with $\overline{\mathcal{E}_1}$. Then we have that

$$
\begin{aligned}
\mathbb{P}\{\overline{\mathcal{E}_1}\} ={}&\mathbb{E}\left[g(S, \{\xi_k\})\right] \\
={}&\mathbb{E}\left[\mathbb{E}\left[g(S, \{\xi_k\}) \mid S_1, \{\xi_k\}\right]\right] \\
={}&\mathbb{E}\left[\mathbb{P}\left\{\max_{k\in[K]}|R(A(S_1, \xi_k)) - R_{S_2}(A(S_1, \xi_k))| \geq M\sqrt{\frac{\log(2K/\delta)}{2\mu N}}\right\} \mid S_1, \{\xi_k\}\right] \\
\overset{\zeta_1}{\leq}{}&\mathbb{E}\left[\frac{\delta}{2} \mid S_1, \{\xi_k\}\right] = \frac{\delta}{2},
\end{aligned}
$$

where in "$\zeta_1$" we have used Hoeffding's inequality and union bound, keeping in mind the independence among $\{A_k\}$, $S_1$ and $S_2$. Further, from Lemma 11 we have $\mathbb{P}\{\overline{\mathcal{E}_2}\} \leq \frac{\delta}{2}$. Combining this and the preceding bound yields

$$
\mathbb{P}\{\mathcal{E}\} \geq \mathbb{P}_{S, \{A_k\}}\{\mathcal{E}_1 \cap \mathcal{E}_2\} \geq 1 - \mathbb{P}\{\overline{\mathcal{E}_1}\} - \mathbb{P}\{\overline{\mathcal{E}_2}\} \geq 1 - \frac{\delta}{2} - \frac{\delta}{2} = 1 - \delta.
$$

The proof is concluded. $\qquad\square$

## C  Proofs for Section 4

In this section, we present the technical proofs for the main results stated in Section 4.

### C.1  Auxiliary lemmas

We need the following lemma from Hardt et al. (2016) which shows that SGD iteration is non-expansive for convex and smooth losses.

**Lemma 12** (Hardt et al. (2016)). *Let $f : \mathcal{W} \mapsto \mathbb{R}$ be a convex and $L$-smooth function. Then for any $w, \tilde{w} \in \mathcal{W}$ and $\alpha \leq 2/L$, we have the following bound holds*

$$
\|w - \alpha\nabla f(w) - (\tilde{w} - \alpha\nabla f(\tilde{w}))\| \leq \|w - \tilde{w}\|.
$$

The following lemma, which can be proved by induction (see, e.g., Schmidt et al., 2011), will be used to prove the main results in Section 4.

**Lemma 13.** *Assume that the nonnegative sequence $\{u_\tau\}_{\tau \geq 1}$ satisfies the following recursion for all $t \geq 1$:*

$$u_t^2 \leq S_t + \sum_{\tau=1}^{t} \alpha_\tau u_\tau,$$

*with $\{S_\tau\}_{\tau \geq 1}$ an increasing sequence, $S_0 \geq u_0^2$ and $\alpha_\tau \geq 0$ for all $\tau$. Then, the following inequality holds for all $t \geq 1$:*

$$u_t \leq \sqrt{S_t} + \sum_{\tau=1}^{t} \alpha_\tau.$$

For analyzing SGD with convex and non-smooth loss functions, we need the following lemma by Bassily et al. (2020, Lemma 3.1) that quantifies the deviation between the online gradient descent trajectories.

**Lemma 14** (Bassily et al. (2020)). *Consider the two sequences $\{w_t\}_{t \geq 0}$ and $\{\tilde{w}_t\}_{t \geq 0}$ generated according to the following recursions respectively over the convex and $G$-Lipschitz objectives $\{f_t\}_{t \geq 0}$ and $\{\tilde{f}_t\}_{t \geq 0}$ from $w_0 = \tilde{w}_0$:*

$$w_t = \Pi_{\mathcal{W}} \left( w_{t-1} - \eta_t \nabla f_{t-1}(w_{t-1}) \right)$$
$$\tilde{w}_t = \Pi_{\mathcal{W}} \left( \tilde{w}_{t-1} - \eta_t \nabla \tilde{f}_{t-1}(\tilde{w}_{t-1}) \right).$$

*Let $t_0 := \inf\{t : f_t \neq f_t'\}$ and $\beta_t := \mathbf{1}_{\{f_t \neq f_t'\}}$. Then for any $T \geq 1$,*

$$\|w_T - \tilde{w}_T\| \leq 2G \sqrt{\sum_{t=t_0}^{T-1} \eta_t^2} + 4G \sum_{t=t_0+1}^{T-1} \eta_t \beta_t.$$

## C.2 Proof of Lemma 1

Here we prove Lemma 1 that establishes the $L_2$-uniform stability of $A_{\text{SGD-w}}$ with convex and smooth loss functions.

*Proof.* Given any pair of data sets $S, \tilde{S}$ that differ in a single element, let us define the sequences $\{w_t\}_{t \in [T]}$ and $\{\tilde{w}_t\}_{t \in [T]}$ that are respectively generated over $S$ and $\tilde{S}$ via $A_{\text{SGD-w}}$ via sample path $\xi = \{i_t\}_{t \in [T]}$. Note by assumption that $w_0 = \tilde{w}_0$. We distinguish the following two complementary cases.

**Case I:** $Z_{i_t} = \tilde{Z}_{i_t}$. In this case, by invoking Lemma 12 we get

$$
\begin{aligned}
\|w_t - \tilde{w}_t\|^2 &= \|\Pi_{\mathcal{W}}(w_{t-1} - \eta_t \nabla_w \ell(w_{t-1}; Z_{i_t})) - \Pi_{\mathcal{W}}(\tilde{w}_{t-1} - \eta_t \nabla_w \ell(\tilde{w}_{t-1}; \tilde{Z}_{i_t}))\|^2 \\
&\leq \|w_{t-1} - \eta_t \nabla_w \ell(w_{t-1}; Z_{i_t}) - (\tilde{w}_{t-1} - \eta_t \nabla_w \ell(\tilde{w}_{t-1}; Z_{i_t}))\|^2 \\
&\leq \|w_{t-1} - \tilde{w}_{t-1}\|^2.
\end{aligned}
\tag{31}
$$

**Case II:** $Z_{i_t} \neq \tilde{Z}_{i_t}$. In this case, we have

$$
\begin{aligned}
\|w_t - \tilde{w}_t\|^2 &= \|\Pi_{\mathcal{W}}(w_{t-1} - \eta_t \nabla_w \ell(w_{t-1}; Z_{i_t})) - \Pi_{\mathcal{W}}(\tilde{w}_{t-1} - \eta_t \nabla_w \ell(\tilde{w}_{t-1}; \tilde{Z}_{i_t}))\|^2 \\
&\leq \|w_{t-1} - \eta_t \nabla_w \ell(w_{t-1}; Z_{i_t}) - (\tilde{w}_{t-1} - \eta_t \nabla_w \ell(\tilde{w}_{t-1}; \tilde{Z}_{i_t}))\|^2 \\
&\leq \left( \|w_{t-1} - \tilde{w}_{t-1}\| + \eta_t (\|\nabla_w \ell(w_{t-1}; Z_{i_t})\| + \|\nabla_w \ell(\tilde{w}_{t-1}; \tilde{Z}_{i_t})\|) \right)^2 \\
&\leq (\|w_{t-1} - \tilde{w}_{t-1}\| + 2G\eta_t)^2 \\
&= \|w_{t-1} - \tilde{w}_{t-1}\|^2 + 4G\eta_t \|w_{t-1} - \tilde{w}_{t-1}\| + 4G^2 \eta_t^2,
\end{aligned}
\tag{32}
$$

where in the last but inequality we have used $\ell(\cdot; \cdot)$ is $G$-Lipschitz with respect to its first argument.

Let $\beta_t = \beta_t(S, \tilde{S}, i_t) := \mathbf{1}_{\{Z_{i_t} \neq \tilde{Z}_{i_t}\}}$ be the random indication function associated with the event $Z_{i_t} \neq \tilde{Z}_{i_t}$. Based on the recursion forms (31) and (32) and the condition $w_0 = \tilde{w}_0$ we can show that

for all $t \geq 1$,

$$\|w_t - \tilde{w}_t\|^2 \leq \sum_{\tau=1}^t 4G\beta_\tau \eta_\tau \|w_{\tau-1} - \tilde{w}_{\tau-1}\| + \sum_{\tau=1}^t 4G^2\beta_\tau \eta_\tau^2.$$

Then applying Lemma 13 with simple algebraic manipulation yields

$$\|w_t - \tilde{w}_t\|^2 \leq 8G^2\left(\sum_{\tau=1}^t \beta_\tau \eta_\tau^2 + 4\left(\sum_{\tau=1}^t \beta_\tau \eta_\tau\right)^2\right).$$

Since by assumption $S$ and $\tilde{S}$ differ only in a single element, under the scheme of uniform sampling without replacement, we can see that $\beta_t \sim \texttt{Bernoulli}(1/N)$ and $\{\beta_t\}_{t \in [T]}$ is an i.i.d. sequence of Bernoulli random variables. It follows that

$$\mathbb{E}\left[\|w_t - \tilde{w}_t\|^2\right]$$

$$\leq 8G^2\left(\sum_{\tau=1}^t \mathbb{E}\left[\beta_\tau\right]\eta_\tau^2 + 4\mathbb{E}\left[\left(\sum_{\tau=1}^t \beta_\tau \eta_\tau\right)^2\right]\right)$$

$$= 8G^2\left(\sum_{\tau=1}^t \mathbb{E}\left[\beta_\tau + 4\beta_\tau^2\right]\eta_\tau^2 + 4\sum_{\tau \neq \tau'} 1\mathbb{E}\left[\beta_\tau \beta_{\tau'}\right]\eta_\tau \eta_{\tau'}\right)$$

$$= 8G^2\left(\frac{5}{N}\sum_{\tau=1}^t \eta_\tau^2 + \frac{4}{N^2}\left(\sum_{\tau=1}^t \eta_\tau\right)^2\right) \leq 40G^2\left(\frac{1}{N}\sum_{\tau=1}^T \eta_\tau^2 + \frac{1}{N^2}\left(\sum_{\tau=1}^T \eta_\tau\right)^2\right),$$

where we have used $\mathbb{E}[\beta_t] = \mathbb{E}[\beta_t^2] = \frac{1}{N}$. The convexity of squared Euclidean norm leads to

$$\mathbb{E}\left[\|\bar{w}_T - \tilde{w}_T\|^2\right] \leq \frac{\sum_{t=1}^T \mathbb{E}\left[\|w_t - \tilde{w}_t\|^2\right]}{T} \leq 40G^2\left(\frac{1}{N}\sum_{t=1}^T \eta_t^2 + \frac{1}{N^2}\left(\sum_{t=1}^T \eta_t\right)^2\right).$$

For each $i \in [N]$, let $\left\{w_t^{(i)}\right\}_{t \in [T]}$ be the sequence generated over $S^{(i)}$ by $A_{\texttt{SGD-w}}$. Since the above holds for any $S \doteq \tilde{S}$, we must have

$$\sup_{S, Z_i'} \mathbb{E}_\xi\left[\left\|\bar{w}_T - \bar{w}_T^{(i)}\right\|^2\right] \leq 40G^2\left(\frac{1}{N}\sum_{t=1}^T \eta_t^2 + \frac{1}{N^2}\left(\sum_{t=1}^T \eta_t\right)^2\right).$$

Finally, since the loss is $G$-Lipschitz, it follows from the above that for all $i \in [N]$,

$$\sup_{S, Z_i', Z} \mathbb{E}_\xi\left[\left(\ell(\bar{w}_T; Z) - \ell(\bar{w}_T^{(i)}; Z)\right)^2\right] \leq 40G^4\left(\frac{1}{N}\sum_{t=1}^T \eta_t^2 + \frac{1}{N^2}\left(\sum_{t=1}^T \eta_t\right)^2\right).$$

This proves the desired $L_2$-uniform stability of algorithm. $\qquad\square$

### C.3 Proof of Lemma 2

In this subsection we prove Lemma 2 that establishes the $L_2$-uniform stability of $A_{\texttt{SGD-w}}$ with convex and non-smooth loss functions.

*Proof.* The proof arguments follow closely those of Lemma 1. Here we reproduce the proof for the sake of completeness. Let us define the sequences $\{w_t\}_{t \in [T]}$ and $\{\tilde{w}_t\}_{t \in [T]}$ that are respectively generated over $S$ and $\tilde{S}$ by $A_{\texttt{SGD-w}}$ via sample path $\xi = \{i_t\}_{t \in [T]}$. Suppose that $S \doteq \tilde{S}$ and consider a hitting time variable $t_0 = \inf\{t : Z_{i_t} \neq \tilde{Z}_{i_t}\}$. Let $\beta_t = \beta_t(S, \tilde{S}, i_t) := \mathbf{1}_{\left\{Z_{i_t} \neq \tilde{Z}_{i_t}\right\}}$ be the random indication function associated with event $Z_{i_t} \neq \tilde{Z}_{i_t}$. Then $\{\beta_t\}_{t \in [T]}$ is an i.i.d. sequence of

Bernoulli$(1/N)$ random variables. Conditioned on $t_0$, it has been shown by Bassily et al. (2020) (see Lemma 14) that

$$\|w_t - \tilde{w}_t\| \leq 2G\sqrt{\sum_{\tau=t_0}^t \eta_\tau^2} + 4G\sum_{\tau=t_0+1}^t \beta_\tau \eta_\tau \leq 2G\sqrt{\sum_{\tau=1}^t \eta_\tau^2} + 4G\sum_{\tau=1}^t \beta_\tau \eta_\tau. \tag{33}$$

Given $S$ and $\tilde{S}$, based on the square of the bound (33) we can show that

$$\mathbb{E}\left[\|w_t - \tilde{w}_t\|^2\right] \leq \mathbb{E}\left[8G^2\sum_{\tau=1}^t \eta_\tau^2 + 32G^2\left(\sum_{\tau=1}^t \beta_\tau \eta_\tau\right)^2\right]$$

$$= 8G^2\sum_{\tau=1}^t \eta_\tau^2 + 32G^2\mathbb{E}\left[\sum_{\tau=1}^t \beta_\tau^2 \eta_\tau^2 + \sum_{\tau\neq\tau'} \beta_\tau \beta_{\tau'} \eta_\tau \eta_{\tau'}\right]$$

$$= 8G^2\sum_{\tau=1}^t \eta_\tau^2 + 32G^2\left(\frac{1}{N}\sum_{\tau=1}^t \eta_\tau^2 + \frac{1}{N^2}\sum_{\tau\neq\tau'} \eta_\tau \eta_{\tau'}\right)$$

$$\leq 40G^2\sum_{\tau=1}^t \eta_\tau^2 + \frac{32G^2}{N^2}\left(\sum_{\tau=1}^t \eta_\tau\right)^2,$$

where we have used $\mathbb{E}[\beta_t] = \mathbb{E}[\beta_t^2] = \frac{1}{N}$. It follows directly from the convexity of squared loss that

$$\mathbb{E}\left[\|\bar{w}_T - \tilde{\bar{w}}_T\|^2\right] \leq 40G^2\sum_{t=1}^T \eta_t^2 + \frac{32G^2}{N^2}\left(\sum_{t=1}^T \eta_t\right)^2.$$

Since the above holds for any pair of $S \doteq \tilde{S}$, we have that for all $i \in [N]$,

$$\sup_{S,Z_i'} \mathbb{E}_\xi\left[\left\|\bar{w}_T - \bar{w}_T^{(i)}\right\|^2\right] \leq 40G^2\sum_{t=1}^T \eta_t^2 + \frac{32G^2}{N^2}\left(\sum_{t=1}^T \eta_t\right)^2,$$

where $\{w_t^{(i)}\}_{t\in[T]}$ is generated over $S^{(i)}$ by $A_{\texttt{SGD-w}}$. Finally, since the loss is $G$-Lipschitz, it follows from the above bound that for all $i \in [N]$,

$$\sup_{S,Z_i',Z} \mathbb{E}_\xi\left[\left(\ell(\bar{w}_T;Z) - \ell(\bar{w}_T^{(i)};Z)\right)^2\right] \leq 40G^4\sum_{t=1}^T \eta_t^2 + \frac{32G^4}{N^2}\left(\sum_{t=1}^T \eta_t\right)^2.$$

This proves the desired $L_2$-uniform stability of algorithm. □

## C.4 Proof of Lemma 3

In this subsection we prove Lemma 3 which establishes the $L_2$-uniform stability of $A_{\texttt{SGD-w}}$ with non-convex and smooth loss functions.

*Proof.* Let us define the sequences $\{w_t\}_{t\in[T]}$ and $\{\tilde{w}_t\}_{t\in[T]}$ that are respectively generated over $S$ and $\tilde{S}$ by $A_{\texttt{SGD-w}}$ via sample path $\xi = \{i_t\}_{t\in[T]}$. Suppose that $S \doteq \tilde{S}$. Let us consider $\Delta_t := \mathbb{E}[\|w_t - \tilde{w}_t\|]$. Then based on the arguments of Hardt et al. (2016, Theorem 3.8) we know that with probability $1 - \frac{1}{N}$ over $i_t$, $\|w_t - \tilde{w}_t\| \leq (1 + \eta_t L)\|w_{t-1} - \tilde{w}_{t-1}\|$, and $\|w_t - \tilde{w}_t\| \leq$

$\|w_{t-1} - \tilde{w}_{t-1}\| + 2G\eta_t$ with probability $\frac{1}{N}$. Therefore we have

$$\Delta_t \leq \left(1 - \frac{1}{N}\right)(1 + \eta_t L)\Delta_{t-1} + \frac{1}{N}(\Delta_{t-1} + 2G\eta_t)$$

$$= \left(\left(1 - \frac{1}{N}\right)(1 + \eta_t L) + \frac{1}{N}\right)\Delta_{t-1} + \frac{2G\eta_t}{N}$$

$$= \left(1 + \left(1 - \frac{1}{N}\right)\eta_t L\right)\Delta_{t-1} + \frac{2G\eta_t}{N}$$

$$\leq \exp\left(\left(1 - \frac{1}{N}\right)\eta_t L\right)\Delta_{t-1} + \frac{2G\eta_t}{N}$$

$$\leq \exp\left(\eta_t L\right)\Delta_{t-1} + \frac{2G\eta_t}{N},$$

where we have used $1 + x \leq \exp(x)$. Then we can unwind the above recursion form to obtain that for all $t \geq 1$,

$$\Delta_t \leq \sum_{\tau=1}^{t} \prod_{i=\tau+1}^{t} \exp\left(\eta_i L\right)\frac{2G\eta_\tau}{N} = \frac{2G}{N}\sum_{\tau=1}^{t} \exp\left(L\sum_{i=\tau+1}^{t}\eta_i\right)\eta_\tau, \tag{34}$$

where we have used $\Delta_0 = 0$. Now we consider $\Gamma_t := \mathbb{E}\left[\|w_t - \tilde{w}_t\|^2\right]$. Then we can verify that with probability $1 - \frac{1}{N}$ over $i_t$, $\|w_t - \tilde{w}_t\|^2 \leq (1 + \eta_t L)^2 \|w_{t-1} - \tilde{w}_{t-1}\|^2$, and with probability $\frac{1}{N}$,

$$\|w_t - \tilde{w}_t\|^2 \leq (\|w_{t-1} - \tilde{w}_{t-1}\| + 2G\eta_t)^2 = \|w_{t-1} - \tilde{w}_{t-1}\|^2 + 4G\eta_t\|w_{t-1} - \tilde{w}_{t-1}\| + 4G^2\eta_t^2.$$

Therefore we have

$$\Gamma_t \leq \left(1 - \frac{1}{N}\right)(1 + \eta_t L)^2\Gamma_{t-1} + \frac{1}{N}\left(\Gamma_{t-1} + 4G\eta_t\Delta_{t-1} + 4G^2\eta_t^2\right)$$

$$\leq \left(\left(1 - \frac{1}{N}\right)(1 + \eta_t L)^2 + \frac{1}{N}\right)\Gamma_{t-1} + \frac{4G^2}{N}\left(\underbrace{\eta_t^2 + 2\eta_t\sum_{\tau=1}^{t-1}\exp\left(L\sum_{i=\tau+1}^{t-1}\eta_i\right)\eta_\tau}_{u_t}\right)$$

$$= \left(1 + \left(1 - \frac{1}{N}\right)(2\eta_t L + \eta_t^2 L^2)\right)\Gamma_{t-1} + \frac{4G^2 u_t}{N}$$

$$\leq \exp\left(\left(1 - \frac{1}{N}\right)(2\eta_t L + \eta_t^2 L^2)\right)\Gamma_{t-1} + \frac{4G^2 u_t}{N}$$

$$\leq \exp\left(2\eta_t L + \eta_t^2 L^2\right)\Gamma_{t-1} + \frac{4G^2 u_t}{N},$$

where in the second inequality we have used the bound (34) on $\Delta_t$. Recall that $\Gamma_0 = 0$. Then we can unwind the above recursion form to obtain

$$\Gamma_t \leq \frac{4G^2}{N}\sum_{\tau=1}^{t}\left\{\prod_{i=\tau+1}^{t}\exp\left(2\eta_i L + \eta_i^2 L^2\right)\right\}u_\tau \leq \frac{4G^2}{N}\sum_{\tau=1}^{t}\exp\left(3L\sum_{i=\tau+1}^{t}\eta_i\right)u_\tau,$$

where we have used $\eta_t \leq 1/L$. It follows immediately from the convexity that

$$\mathbb{E}\left[\|\bar{w}_T - \tilde{\bar{w}}_T\|^2\right] \leq \frac{\sum_{t=1}^{T}\mathbb{E}\left[\|w_t - \tilde{w}_t\|^2\right]}{T} \leq \frac{4G^2}{N}\sum_{t=1}^{T}\exp\left(3L\sum_{\tau=t+1}^{T}\eta_\tau\right)u_t.$$

Since the above holds for any $S \doteq \tilde{S}$, we have that for all $i \in [N]$,

$$\sup_{S,Z_i'}\mathbb{E}_\xi\left[\left\|\bar{w}_T - \bar{w}_T^{(i)}\right\|^2\right] \leq \frac{4G^2}{N}\sum_{t=1}^{T}\exp\left(3L\sum_{\tau=t+1}^{T}\eta_\tau\right)u_t,$$

---

**Algorithm 3:** $A_{\texttt{SGD-w/o}}$: SGD under Without-Replacement Sampling

---
**Input** : Data set $S = \{Z_i\}_{i \in [N]}$, step-sizes $\{\eta_t\}_{t \geq 1}$, #iterations $T$, initialization $w_0$.
**Output** : $\bar{w}_T = \frac{1}{T} \sum_{t \in [T]} w_t$.
**for** $t = 1, 2, ..., T$ **do**
  | Uniformly randomly sample an index $i_t \in [N]$ *without* replacement;
  | Compute $w_t = \Pi_{\mathcal{W}} (w_{t-1} - \eta_t \nabla_w \ell(w_{t-1}; Z_{i_t}))$.
**end**

---

where $\{w_t^{(i)}\}_{t \in [T]}$ is generated over $S^{(i)}$ by $A_{\texttt{SGD-w}}$. Finally, since the loss is $G$-Lipschitz, it follows from the above that for all $i \in [N]$,

$$\sup_{S, Z_i', Z} \mathbb{E}_{\xi} \left[ \left( \ell(\bar{w}_T; Z) - \ell(\bar{w}_T^{(i)}; Z) \right)^2 \right] \leq \frac{4G^4}{N} \sum_{t=1}^{T} \exp \left( 3L \sum_{\tau=t+1}^{T} \eta_\tau \right) u_t.$$

This proves the desired $L_2$-uniform stability of algorithm. $\qquad \square$

## D  Augmented Results for SGD under Without-Replacement Sampling

In this section, we further consider applying our main results in Theorem 2 to the variant of SGD under without-replacement sampling ($A_{\texttt{SGD-w/o}}$), as is outlined in Algorithm 3. For the sake of simplicity and readability, we only consider single-epoch processing with $T \leq N$. The extensions of our analysis to multi-epoch processing, i.e., $T \leq rN$ for some integer $r \geq 1$ are more or less straightforward and thus the details are omitted.

### D.1  Results for convex and smooth loss

We start by considering the regime where the loss function is convex and smooth. We need the following lemma on the $L_2$-uniform stability of $A_{\texttt{SGD-w/o}}$ which can be proved based on the result from Bassily et al. (2020, Lemma 3.1).

**Lemma 15.** *Suppose that the loss function $\ell(\cdot; \cdot)$ is convex, $G$-Lipschitz and $L$-smooth with respect to its first argument. Assume that $\eta_t \leq 2/L$ for all $t \geq 1$. Consider $T \leq N$. Then $A_{SGD\text{-}w/o}$ has $L_2$-uniform stability with parameter*

$$\gamma_{L_2, N} = 2G^2 \sqrt{\frac{1}{N} \sum_{t=1}^{T} \eta_t^2}.$$

*Proof.* For any fixed pair of data sets $S, \tilde{S}$ that differ in a single element, let us define the sequences $\{w_t\}_{t \in [T]}$ and $\{\tilde{w}_t\}_{t \in [T]}$ that are respectively generated over $S$ and $\tilde{S}$ by $A_{\texttt{SGD-w/o}}$ via sample path $\xi = \{i_t\}_{t \in [T]}$. Recall that $T \leq N$. Let us define a stopping time variable $t_0$ such that $Z_{\xi_{t_0}} \neq \tilde{Z}_{\xi_{t_0}}$. Since $S \doteq \tilde{S}$, the uniform randomness of $i_t$ implies that

$$\mathbb{P}(t_0 = j) = \frac{1}{N}, \quad j \in [N].$$

In the proof of Corollary 1 we have already shown that $\|w_t - \tilde{w}_t\|^2 \leq \|w_{t-1} - \tilde{w}_{t-1}\|^2$ if $Z_{i_t} = \tilde{Z}_{i_t}$ and $\|w_t - \tilde{w}_t\|^2 \leq \|w_{t-1} - \tilde{w}_{t-1}\|^2 + 4G\eta_t \|w_{t-1} - \tilde{w}_{t-1}\| + 4G^2 \eta_t^2$ otherwise. Therefore, the without-replacement sampling implies that the following bound holds for any given $t_0 \leq t \leq T$:

$$\|w_t - \tilde{w}_t\|^2 \leq 4G^2 \eta_{t_0}^2,$$

and $\|w_t - \tilde{w}_t\|^2 = 0$ for $0 \leq t < t_0$. Then based on the law of total expectation we can show that

$$\mathbb{E}\left[\|w_t - \tilde{w}_t\|^2\right] \leq \frac{4G^2}{N} \sum_{t_0=1}^{t} \eta_{t_0}^2 \leq \frac{4G^2}{N} \sum_{t=1}^{T} \eta_t^2.$$

The convexity of squared Euclidean norm leads to

$$\mathbb{E}\left[\|\bar{w}_T - \tilde{\bar{w}}_T\|^2\right] \leq \frac{\sum_{t=1}^T \mathbb{E}\left[\|w_t - \tilde{w}_t\|^2\right]}{T} \leq \frac{4G^2}{N}\sum_{t=1}^T \eta_t^2.$$

Since the above holds for any $S \doteq \tilde{S}$, we have that for all $i \in [N]$,

$$\sup_{S,Z_i'} \mathbb{E}_\xi\left[\left\|\bar{w}_T - \bar{w}_T^{(i)}\right\|^2\right] \leq \frac{4G^2}{N}\sum_{t=1}^T \eta_t^2,$$

where $\{w_t^{(i)}\}_{t\in[T]}$ is generated over $S^{(i)}$ by $A_{\text{SGD-w/o}}$. Finally, since the loss is $G$-Lipschitz, it follows from the above bound that for all $i \in [N]$,

$$\sup_{S,Z_i',Z} \mathbb{E}_\xi\left[\left(\ell(\bar{w}_T; Z) - \ell(\bar{w}_T^{(i)}; Z)\right)^2\right] \leq \frac{4G^4}{N}\sum_{t=1}^T \eta_t^2.$$

This proves the desired $L_2$-uniform stability of algorithm. $\qquad\square$

The following result is a direct consequence of Theorem 2 when invoking Algorithm 1 to $A_{\text{SGD-w/o}}$ with convex and smooth loss.

**Corollary 4.** *Suppose that the loss function $\ell(\cdot;\cdot) \in [0, M]$ is convex, $G$-Lipschitz and $L$-smooth with respect to its first argument. Consider Algorithm 1 specified to $A_{\text{SGD-w/o}}$ with $T = N$ and learning rate $\eta_t \leq 2/L$ for all $t \geq 1$. Then for any $\delta \in (0,1)$ and $K \geq 2\log(\frac{2}{\delta})$, with probability at least $1 - \delta$ over the randomness of $S$ and $\{\xi\}_{k\in[K]}$, the generalization bound of Algorithm 1 is upper bounded as*

$$|R(A_{\text{SGD-w/o}}(S_1, \xi_{k^*})) - R_S(A_{\text{SGD-w/o}}(S_1, \xi_{k^*}))|$$
$$\lesssim G^2 \log(N) \log\left(\frac{1}{\delta}\right)\sqrt{\frac{1}{(1-\mu)N}\sum_{t=1}^N \eta_t^2} + \frac{M}{\sqrt{\mu(1-\mu)}}\sqrt{\frac{\log(K/\delta)}{N}}.$$

*Proof.* For the considered convex and smooth losses, from Lemma 15 we know that $A_{\text{SGD-w/o}}$ with $T = N$ iterations has $L_2$-uniform stability with parameter $\gamma_{L_2,N} = 2G^2\sqrt{\frac{1}{N}\sum_{t=1}^N \eta_t^2}$. Then the generalization error bound follows immediately from Theorem 2. $\qquad\square$

**Remark 12.** *Specially for constant learning rates $\eta_t \equiv \eta \asymp \frac{1}{\sqrt{N}}$ and $K \asymp \log\left(\frac{1}{\delta}\right)$, Corollary 4 admits a high-probability generalization bound of order $\mathcal{O}\left(\sqrt{\frac{\log(1/\delta)}{N}} + \frac{\log(N)\log(1/\delta)}{\sqrt{N}}\right)$. For time varying learning rates $\eta_t \asymp \frac{1}{\sqrt{t}}$, the generalization bound scales as $\mathcal{O}\left(\frac{\log(N)\log(1/\delta)}{\sqrt{N}} + \sqrt{\frac{\log(1/\delta)}{N}}\right)$.*

Further assume that $\mathcal{W}$ is bounded with diameter $D$. Consider the constant learning rate $\eta_t \equiv \min\{\frac{2}{L}, \frac{D}{G\sqrt{N}}\}$. Then the following in-expectation optimization error bound of $A_{\text{SGD-w/o}}$ with convex and smooth loss functions is known (Nagaraj et al., 2019, Theorem 3):

$$\mathbb{E}_\xi\left[R_S(\bar{w}_T) - \min_{w\in\mathcal{W}} R_S(w)\right] \lesssim \frac{D^2 L}{N} + \frac{GD}{\sqrt{N}}.$$

Invoking generic bound (14) combined with Lemma 15 and the above sub-optimality bound yields the following excess risk bound of (modified) Algorithm 1:

$$R(A_{\text{SGD-w/o}}(S_1, \xi_{k^*})) - R^* \lesssim G^2 \log(N)\log\left(\frac{1}{\delta}\right)\sqrt{\frac{1}{(1-\mu)N}\sum_{t=1}^N \eta_t^2} + \frac{M}{\sqrt{\mu(1-\mu)}}\sqrt{\frac{\log(K/\delta)}{N}}$$
$$+ \frac{GDK}{\sqrt{(1-\mu)N}} + \frac{D^2 L}{(1-\mu)N}.$$

### D.2 Results for convex and non-smooth loss

We now turn to study the case of convex and non-smooth losses. The following lemma is about the $L_2$-uniform stability of $A_{\texttt{SGD-w/o}}$ in this case.

**Lemma 16.** *Suppose that the loss function $\ell(\cdot; \cdot)$ is convex and $G$-Lipschitz with respect to its first argument. Consider $T \leq N$. Then $A_{\texttt{SGD-w/o}}$ has $L_2$-uniform stability with parameter*

$$\gamma_{L_2, N} = 2G^2 \sqrt{\frac{1}{N} \sum_{t_0=1}^{T} \sum_{t=t_0}^{T} \eta_t^2}.$$

*Proof.* The proof arguments are similar to those of Lemma 15. Here we reproduce the proof for the sake of completeness. For any fixed pair of data sets $S, \tilde{S}$ that differ in a single element, let us define the sequences $\{w_t\}_{t \in [T]}$ and $\{\tilde{w}_t\}_{t \in [T]}$ that are respectively generated over $S$ and $\tilde{S}$ by $A_{\texttt{SGD-w/o}}$ via sample path $\xi = \{i_t\}_{t \in [T]}$. Recall that $T \leq N$. Let us define a stopping time variable $t_0$ such that $Z_{\xi_{t_0}} \neq \tilde{z}_{\xi_{t_0}}$. Since $S \doteq \tilde{S}$, the uniform randomness of $i_t$ and the without-replacement sampling strategy yield

$$\mathbb{P}(t_0 = j) = \frac{1}{N}, \quad j \in [N].$$

For any $t_0 \leq t \leq T$, under without-replacement sampling, it follows from Lemma 14 that

$$\|w_t - \tilde{w}_t\|^2 \leq 4G^2 \sum_{\tau=t_0}^{t} \eta_\tau^2.$$

We use the convention $\sum_{\tau=t_0}^{t} \eta_\tau^2 = 0$ for $0 \leq t < t_0$. Then according to the law of total expectation we must have

$$\mathbb{E}\left[\|w_t - \tilde{w}_t\|^2\right] \leq \frac{4G^2}{N} \sum_{t_0=1}^{t} \sum_{\tau=t_0}^{t} \eta_\tau^2 \leq \frac{4G^2}{N} \sum_{t_0=1}^{T} \sum_{\tau=t_0}^{T} \eta_\tau^2.$$

The convexity of squared Euclidean norm leads to

$$\mathbb{E}\left[\|\bar{w}_T - \tilde{\bar{w}}_T\|^2\right] \leq \frac{\sum_{t=1}^{T} \mathbb{E}\left[\|w_t - \tilde{w}_t\|^2\right]}{T} \leq \frac{4G^2}{N} \sum_{t_0=1}^{T} \sum_{t=t_0}^{T} \eta_t^2.$$

Since the above holds for any $S \doteq \tilde{S}$, the following bound holds for all $i \in [N]$:

$$\sup_{S, Z_i'} \mathbb{E}_\xi \left[\left\|\bar{w}_T - \bar{w}_T^{(i)}\right\|^2\right] \leq \frac{4G^2}{N} \sum_{t_0=1}^{T} \sum_{t=t_0}^{T} \eta_t^2,$$

where $\{w_t^{(i)}\}_{t \in [T]}$ is generated over $S^{(i)}$ by $A_{\texttt{SGD-w/o}}$. Finally, since the loss is $G$-Lipschitz, it follows from the above bound that for all $i \in [N]$,

$$\sup_{S, Z_i', Z} \mathbb{E}_\xi \left[\left(\ell(\bar{w}_T; Z) - \ell(\bar{w}_T^{(i)}; Z)\right)^2\right] \leq \frac{4G^4}{N} \sum_{t_0=1}^{T} \sum_{t=t_0}^{T} \eta_t^2.$$

This proves the desired $L_2$-uniform stability of algorithm. $\qquad \square$

With the above lemma in hand, we can establish the following result as a direct consequence of Theorem 2 when invoking Algorithm 1 to $A_{\texttt{SGD-w/o}}$ with convex and non-smooth losses.

**Corollary 5.** *Suppose that the loss function $\ell(\cdot; \cdot)$ is convex and $G$-Lipschitz with respect to its first argument, and it is bounded in the interval $[0, M]$. Consider Algorithm 1 specified to $A_{\texttt{SGD-w/o}}$ with $T = N$. Then for any $\delta \in (0, 1)$ and $K \geq 2\log(\frac{2}{\delta})$, with probability at least $1 - \delta$ over the randomness of $S$ and $\{\xi_k\}_{k \in [K]}$, the output of Algorithm 1 satisfies*

$$|R(A_{\texttt{SGD-w/o}}(S_1, \xi_{k^*})) - R_S(A_{\texttt{SGD-w/o}}(S_1, \xi_{k^*}))|$$

$$\lesssim G^2 \log(N) \log\left(\frac{1}{\delta}\right) \sqrt{\frac{1}{(1-\mu)N} \sum_{t_0=1}^{N} \sum_{t=t_0}^{N} \eta_t^2} + \frac{M}{\sqrt{\mu(1-\mu)}} \sqrt{\frac{\log(K/\delta)}{N}}.$$

*Proof.* For the considered convex and Lipschitz loss functions, from Lemma 16 we know that $A_{\texttt{SGD-w/o}}$ with $T = N$ iterations has $L_2$-uniform stability by $\gamma_{\texttt{L}_2,N} = 2G^2\sqrt{\frac{1}{N}\sum_{t_0=1}^{N}\sum_{t=t_0}^{N}\eta_t^2}$. The results then follow immediately via invoking Theorem 2 to the considered setting. $\square$

**Remark 13.** *Specially for constant learning rates $\eta_t \equiv \eta$ and setting $K \asymp \log\left(\frac{1}{\delta}\right)$, Corollary 5 admits a high-probability generalization bound of scale $\mathcal{O}\left(\eta\sqrt{N} + \sqrt{\frac{\log(1/\delta)}{N}}\right)$. For time decaying learning rates $\eta_t \asymp \frac{1}{t}$, the generalization bound scales as $\mathcal{O}\left(\sqrt{\frac{\log(N)}{N}} + \sqrt{\frac{\log(1/\delta)}{N}}\right)$.*

Regarding the excess risk bound, under the conditions of Corollary 5 and $K \asymp \log\left(\frac{1}{\delta}\right)$, the risk bound (14) combined with Lemma 16 yields the following exponential risk bound of (modified) Algorithm 1:

$$R(A_{\texttt{SGD-w/o}}(S_1, \xi_{k^*})) - R^*$$

$$\lesssim \Delta_{\text{opt}} + G^2 \log(N)\log\left(\frac{1}{\delta}\right)\sqrt{\frac{1}{(1-\mu)N}\sum_{t_0=1}^{N}\sum_{t=t_0}^{N}\eta_t^2} + \frac{M}{\sqrt{\mu(1-\mu)}}\sqrt{\frac{\log(1/\delta)}{N}}.$$

In the special case of bounded-norm generalized linear models, Shamir (2016) established an in-expectation empirical risk sub-optimality bound $\Delta_{\text{opt}} \lesssim \frac{1}{\sqrt{N}}$ under suitable learning rates. For generic convex and non-smooth losses, however, it still remains unclear to us if similar sub-optimality bounds are available for SGD under without-replacement sampling.

### D.3 Results for non-convex and smooth loss

Finally, we study the performance of Algorithm 1 for $A_{\texttt{SGD-w/o}}$ on smooth but not necessarily convex loss functions. We first establish the following lemma on the $L_2$-uniform stability of $A_{\texttt{SGD-w/o}}$ in the considered non-convex problem setting.

**Lemma 17.** *Suppose that the loss function $\ell(\cdot;\cdot)$ is $G$-Lipschitz and $L$-smooth with respect to its first argument. Consider $T \leq N$. Then $A_{\texttt{SGD-w/o}}$ has $L_2$-uniform stability with parameter*

$$\gamma_{L_2,N} = 2G^2\sqrt{\frac{1}{N}\sum_{t=1}^{T}\exp\left(2L\sum_{\tau=t+1}^{T}\eta_\tau\right)\eta_t^2}.$$

*Proof.* Let us define the sequences $\{w_t\}_{t\in[T]}$ and $\{\tilde{w}_t\}_{t\in[T]}$ that are respectively generated over $S$ and $\tilde{S}$ by $A_{\texttt{SGD-w/o}}$ via sample path $\xi = \{i_t\}_{t\in[T]}$. Suppose that $S \doteq \tilde{S}$. Recall that $T \leq N$. Let us define a stopping time variable $t_0$ such that $Z_{\xi_{t_0}} \neq \tilde{Z}_{\xi_{t_0}}$. Since $S \doteq \tilde{S}$, the without-replacement sampling implies that

$$\mathbb{P}(t_0 = j) = \frac{1}{N}, \quad j \in [N].$$

In view of Lemma 12 we know that $\|w_t - \tilde{w}_t\| \leq (1+\eta_t L)\|w_{t-1} - \tilde{w}_{t-1}\|$ if $Z_{i_t} = Z'_{i_t}$, and $\|w_{t_0} - \tilde{w}_{t_0}\| \leq 2G\eta_{t_0}$ otherwise due to the assumption that the loss is $G$-Lipschitz. Therefore, it can be directly verified that the following holds for any $t_0 \leq t \leq T$:

$$\|w_t - \tilde{w}_t\|^2 \leq 4G^2\prod_{\tau=t_0+1}^{t}(1+\eta_\tau L)^2\eta_{t_0}^2,$$

where we have used $\prod_{\tau=t_0+1}^{t}(1+\eta_\tau L)^2 = 1$ for $t = t_0$. For $t < t_0$, it is trivial that $\|w_t - \tilde{w}_t\| = 0$. Therefore the law of total expectation yields

$$\mathbb{E}\left[\|w_t - \tilde{w}_t\|^2\right] \leq \frac{4G^2}{N}\sum_{t_0=1}^{t}\prod_{\tau=t_0+1}^{t}(1+\eta_\tau L)^2\eta_{t_0}^2$$

$$\leq \frac{4G^2}{N}\sum_{t_0=1}^{t}\prod_{\tau=t_0+1}^{t}\exp(2\eta_\tau L)\eta_{t_0}^2$$

$$\leq \frac{4G^2}{N}\sum_{t_0=1}^{T}\exp\left(2L\sum_{\tau=t_0+1}^{T}\eta_\tau\right)\eta_{t_0}^2,$$

where we have used $1 + x \leq \exp(x)$. The convexity of Euclidean norm leads to

$$\mathbb{E}\left[\|\bar{w}_T - \tilde{w}_T\|\right] \leq \frac{4G^2}{N} \sum_{t_0=1}^{T} \exp\left(2L \sum_{\tau=t_0+1}^{T} \eta_\tau\right) \eta_{t_0}^2.$$

Since the above holds for any $S \doteq \tilde{S}$, the following bound holds for all $i \in [N]$:

$$\sup_{S, Z_i'} \mathbb{E}_\xi\left[\left\|\bar{w}_T - \bar{w}_T^{(i)}\right\|^2\right] \leq \frac{4G^2}{N} \sum_{t_0=1}^{T} \exp\left(2L \sum_{\tau=t_0+1}^{T} \eta_\tau\right) \eta_{t_0}^2,$$

where $\{w_t^{(i)}\}_{t \in [T]}$ is generated over $S^{(i)}$ by $A_{\texttt{SGD-w/o}}$. Finally, since the loss is $G$-Lipschitz, it follows from the above bound that for all $i \in [N]$,

$$\sup_{S, Z_i', Z} \mathbb{E}_\xi\left[\left(\ell(\bar{w}_T; Z) - \ell(\bar{w}_T^{(i)}; Z)\right)^2\right] \leq \frac{4G^4}{N} \sum_{t_0=1}^{T} \exp\left(2L \sum_{\tau=t_0+1}^{T} \eta_\tau\right) \eta_{t_0}^2.$$

This proves the desired $L_2$-uniform stability of algorithm. $\qquad\square$

With Lemma 17 in place, we can readily derive the following result as a direct application of Theorem 2 to $A_{\texttt{SGD-w/o}}$ with Lipschitz and smooth losses.

**Corollary 6.** *Suppose that the loss function $\ell(\cdot; \cdot)$ is $G$-Lipschitz and $L$-smooth with respect to its first argument, and it is bounded in the interval $[0, M]$. Consider Algorithm 1 specified to $A_{\texttt{SGD-w/o}}$ with $T = N$. Then for any $\delta \in (0, 1)$ and $K \geq 2\log(\frac{2}{\delta})$, with probability at least $1 - \delta$ over the randomness of $S$ and $\{\xi_k\}_{k \in [K]}$, the output of Algorithm 1 satisfies*

$$|R(A_{\texttt{SGD-w/o}}(S_1, \xi_{k^*})) - R_S(A_{\texttt{SGD-w/o}}(S_1, \xi_{k^*}))|$$

$$\lesssim G^2 \log(N) \log\left(\frac{1}{\delta}\right) \sqrt{\frac{1}{(1-\mu)N} \sum_{t=1}^{N} \exp\left(L \sum_{\tau=t+1}^{N} \eta_\tau\right) \eta_t^2} + \frac{M}{\sqrt{\mu(1-\mu)}} \sqrt{\frac{\log(K/\delta)}{N}}.$$

*Proof.* For the considered smooth and Lipschitz loss functions, from Lemma 17 we know that $A_{\texttt{SGD-w/o}}$ with $T = N$ rounds of iteration has $L_2$-uniform stability with parameter $\gamma_{\text{L}_2, N} = 2G^2 \sqrt{\frac{1}{N} \sum_{t=1}^{T} \exp\left(2L \sum_{\tau=t+1}^{T} \eta_\tau\right) \eta_t^2}$. The desired results then follow immediately via invoking Theorem 2 to the considered problem regime. $\qquad\square$

**Remark 14.** *For $K \asymp \log\left(\frac{1}{\delta}\right)$ and the choice of constant learning rates $\eta_t \equiv \frac{1}{LN}$, Corollary 6 admits high-probability generalization bounds of scale $\mathcal{O}\left(\frac{\log(N)\log(1/\delta)}{N} + \sqrt{\frac{\log(1/\delta)}{N}}\right)$. For the choice of time decaying learning rates $\eta_t = \frac{1}{L\nu t}$ with arbitrary $\nu > 2$, it can be verified that the corresponding generalization bound is of scale $\mathcal{O}\left(\frac{\log(N)\log(1/\delta)}{\nu N^{1/2 - 1/\nu}} + \sqrt{\frac{\log(1/\delta)}{N}}\right)$.*

# E   Some Additional Related Work

The idea of using stability of a learning algorithm, namely the sensitivity of estimated model to the changes in training data, for generalization performance analysis dates back to the seventies (Vapnik and Chervonenkis, 1974; Rogers and Wagner, 1978; Devroye and Wagner, 1979). For deterministic learning algorithms, algorithmic stability has been extensively studied with a bunch of applications to establishing strong generalization and excess risk bounds for stable learning models like $k$-NN and regularized ERMs (Bousquet and Elisseeff, 2002; Zhang, 2003; Klochkov and Zhivotovskiy, 2021; Yuan and Li, 2023). The stability theory for randomized learning algorithms was formally introduced and investigated by Elisseeff et al. (2005). In the celebrated work of Hardt et al. (2016), it was shown in that the solution obtained via stochastic gradient descent is expected to be stable and generalize well for smooth convex and non-convex loss functions. For non-smooth convex losses, the stability induced generalization bounds of SGD have been established in expectation (Lei and Ying, 2020) or deviation (Bassily et al., 2020). In the work of Kuzborskij and Lampert (2018), a set

of data-dependent generalization bounds for SGD were derived based on the stability of algorithm. More broadly, generalization bounds for stable learning algorithms that converge to global minima were established in Charles and Papailiopoulos (2018); Lei and Ying (2021). For non-convex sparse learning, algorithmic stability theory has been applied to derive the generalization bounds of the popularly used iterative hard thresholding (IHT) algorithm (Yuan and Li, 2022). The uniform stability bounds on SGD have also been extensively used for designing differential privacy stochastic optimization algorithms (Bassily et al., 2019; Feldman et al., 2020).

The confidence-boosting technique has long been applied for obtaining sharp high-probability excess risk bounds from the corresponding in-expectation bounds (Shalev-Shwartz et al., 2010; Mehta, 2017; Holland, 2021). For generic statistical learning problems, confidence-boosting has been used to convert any low-confidence learning algorithm with linear dependence on $1/\delta$ to a high-confidence algorithm with logarithmic factor $\log(1/\delta)$. For learning with exp-concave losses, a relevant ERM estimator with in-expectation fast rate of convergence was converted to a high-confidence learning algorithm with an almost identical fast rate of convergence up to a logarithmic factor on $1/\delta$ (Mehta, 2017). While sharing a similar spirit, our generalization analysis is substantially more challenging than the existing excess risk analysis in the sense that deriving a favorable first-moment generalization bound for $L_2$-uniformly stable randomized algorithms is highly non-trivial in itself. Bagging (or bootstrap aggregating) is one of the earliest yet most popular ensemble methods that has been widely applied to reduce the variance for unstable learning algorithms such as decision tree and neural networks (Breiman, 1996; Opitz and Maclin, 1999), and sometimes stable algorithms such as SVMs (Valentini and Dietterich, 2003). As an important variant of bagging, subbagging has been proposed to reduce the computational cost of bagging via training base models under without-replacement sampling (Bühlmann, 2012). The stability and generalization bounds of bagging have been analyzed for both uniform (Elisseeff et al., 2005) and non-uniform (Foster et al., 2019) averaging schemes. Unlike these prior results for bagging with averaging aggregation, our confidence-boosting bounds are obtained based on a greedy aggregation scheme which turns out to yield sharper dependence on the stability parameters.