# OpenReview forum: "$L_2$-Uniform Stability of Randomized Learning Algorithms: Sharper Generalization Bounds and Confidence Boosting"
_NeurIPS.cc/2023/Conference — NeurIPS 2023 poster_

### Official Review · Reviewer_7EkU · 2023-07-03

**Soundness:** 2 fair
**Presentation:** 2 fair
**Contribution:** 2 fair
**Rating:** 5
**Confidence:** 3

**Summary:**

This paper considers generalization bounds for SGD with a notion of randomization on learning algorithms. The paper considers a $L_2$-uniform stability framework and establishes generalization inequalities under such a framework. Additional algorithms and analysis using confidence-boosting are discussed.

**Strengths:**

The paper's strengths are summarized in the following criteria.

Originality: The main comparisons made within the paper focus on comparisons against prior literature established for "on-average" uniform stability and "high probability" uniform stability. While SGD's under these two forms of uniform stability have been studied extensively, the authors identify an interesting gap in the literature by looking at "$L_2$" uniform stability.

Quality and clarity: The paper covers a good amount of motivation and background of SGD theory. Some additional comments regarding quality and clarity can be found in latter sections.

Significance: Considering that SGD is broadly used in many areas of ML, understanding more about the theory of SGD is quite important. From this perspective, this paper is relevant to the NeurIPS community.

**Weaknesses:**

While the paper covers good amount of material, one of the weaknesses of the paper is its clarity and structure. There were certain parts of the paper that were not easily readable and confusing. It may be helpful to add in formal definitions and clear up typos/confusions:

-  For example, it may be helpful to elaborate in the paper more on specific definitions such as $\mathcal{Z}$, $\mathcal{R}$, etc.
- Another example is the definition of the parameter $\gamma_{L_2,N}$ in lines 162-164; the words "for every $N\geq 1$" is likely redundant and adds confusion for the reader.

Regarding the main results, considering that an $L_2$ assumption is stronger than an $L_1$ assumption, it does not necessarily make sense to compare Theorem 1 and (4) directly. A better "comparison" may be to derive the optimal ordering for the $L_2$ case, i.e., the corresponding lower bound. It is not immediately clear if the known lower bounds from earlier works with different assumptions can be applied to show that Theorem 1 is tight up to a logarithmic factor apart for some special cases (such as when $A$ is deterministic). Is this analysis available?

**Questions:**

Please refer to the questions listed or implied in the above sections. Additional questions / comments are:

1. Why is the upper bound in Theorem 2 increasing as $K$ increases? One would think that running an algorithm multiple times and selecting the "best" output would yield a "tighter" result. Can this be elaborated?

2. Theorems 1 and 2 appear to be "sub-exponential" bounds with respect to dependency of $\log(1/\delta)$. Is it possible to derive with similar methodologies a "sub-gaussian" type bound?

3. Are there matching lower bounds (perhaps up to logarithmic factors) under the $L_2$ uniform stability assumption?

Below are some typos that may need revision:

- the statement in Theorem 2 has an additional period in line 262.
- "Jensen's ineuqality" in line 164

**Limitations:**

Limitations are not addressed in detail.

---

> ### Author Rebuttal · Authors · 2023-08-09
>
> Thank you for your insightful review and detailed comments for improvement. We sincerely hope that the main concerns raised in the review can be clarified by the following responses.
>
> > **Your comment:** It may be helpful to add in formal definitions and clear up typos/confusions.
>
> **Our response:** Thanks for your very careful inspection of our manuscript. Per your suggestion, we will further clarify notation and fix typos/confusions to improve readability.
>
> > **Your comment:** It is not immediately clear if the known lower bounds from earlier works with different assumptions can be applied to show that Theorem 1 is tight up to a logarithmic factor apart for some special cases (such as when $A$  is deterministic). Is this analysis available?
>
> **Our response:** Yes, we indeed believe that the lower bound analysis of Bousquet et al. [2020, Proposition 9] for sum of functions of Rademacher signs could be extended to sum of functions of Rademacher signs with auxiliary random bits. As an initial thought, one may consider to construct randomized functions of form $g_i(S,\xi)=MZ_i + \frac{\beta}{2}Z_i\left(\sum_{j\neq i}Z_j + \xi_1\right)\xi_2$ where $Z_i$ and $\xi_1$, $\xi_2$ are all Rademacher signs. It can be verified that these $g_i(S,\xi)$ satisfy the L2-bounded-difference property. Consequently, the arguments of Bousquet et al. [2020, Proposition 9] can be naturally adapted to the considered randomized functions to get the desired lower bound that matches the upper bound presented in Proposition 2 (which then implies the main result in Theorem 1). Once ready, we will add the lower bound analysis to the next version.
>
>
> > **Your comment:** Why is the upper bound in Theorem 2 increasing as $K$ increases? One would think that running an algorithm multiple times and selecting the "best" output would yield a "tighter" result. Can this be elaborated?
>
> **Our response:** We would like to clarify that the $\log (K/\delta)$ factor in the bound of Theorem 2 originally arises from the the uniform  generalization bound defined in $\mathcal{E_1}$ (Line #600 of the supplement file), which represents the overhead of simultaneously bounding the generalization performance of $K$ different candidate solutions $\{A(S_1,\xi_k)\}_{k\in [K]}$ over the holdout validation set $S_2$. We will add some remarks on this point in the updated draft.
>
>
> > **Your comment:**  Theorems 1 and 2 appear to be "sub-exponential" bounds with respect to dependency of $\log(1/\delta)$. Is it possible to derive with similar methodologies a "sub-gaussian" type bound?
>
> **Our response:** It is an interesting direction to explore the possibility of deriving exponential generalization bounds of sub-gaussian form. The current sample-splitting technique based on moments concentration inequalities seems not allowing us to remove the sub-exponential terms. We conjecture that a potential workaround is using the the exponential versions of Efron-Stein inequality [Boucheron et al., 2003] instead of the moments counterpart inside the sample-splitting framework. We leave the improvement over poly-logarithmic terms for future investigation.
>
>
> ## References:
>
> Olivier Bousquet, Yegor Klochkov, and Nikita Zhivotovskiy. Sharper bounds for uniformly stable
> algorithms. *Conference on Learning Theory*, pp. 610–626, 2020.
>
> Stephane Boucheron, Gabor Lugosi, and Pascal Massart. Concentration inequalities using the entropy
> method. *Annals of Probability*, 31(3):1583–1614, 2003.

---

> > ### Comment · Reviewer_7EkU · 2023-08-18
> > **Response to rebuttal**
> >
> > Thank you for addressing some of the concerns raised. Please consider adding these discussions with more details in the paper. In terms of clarity of the paper, I would suggest further, detailed, revisions. As also pointed out by other reviewers, there are certain parts of the paper that can be clarified further. I am adjusting my score based on the discussions here, other reviews, and the current state of the paper.

---

> > > ### Author Response · Authors · 2023-08-19
> > > **Thank you for your response**
> > >
> > > We sincerely thank the reviewer for the updated rating score and would like to express our gratitude for your willingness to accept our paper. Per your suggestion, we will be happy to carefully revise the paper to improve its clarity and structure in the mentioned pivotal aspects.

---

### Official Review · Reviewer_RTFd · 2023-07-05

**Soundness:** 3 good
**Presentation:** 3 good
**Contribution:** 2 fair
**Rating:** 6
**Confidence:** 4

**Summary:**

This paper develops new and potentially sharper (as compared to the state of the art) data and randomness dependent (high probability type) generalization bounds using a kind of new and more restrictive notion of algorithmic stability, which the authors call L2-uniform stability. The focus is in particular on generalization analysis for randomized algorithms.

The main results of the paper are then leveraged to derive very detailed high probability generalization bounds for a certain bagging-based meta algorithmic framework. Those results are subsequently instantiated for SGD, in the same boosting-type scheme.

**Strengths:**

The paper is well-written and considers an important and challenging question, namely, that of characterizing the generalization performance of randomized algorithms in a "high-probability" scheme, which is very much attractive especially from a practical/application perspective.

Section 1 (Intro) are particularly well-written and generally a pleasure to read, and well-suited even for people who are not very familiar with algorithmic generalization.

Propositions 1 and 2 are interesting, and could be potentially of independent interest. This is a plus.

Theorem 1, one of the main and "backbone" results, is interesting though somewhat expected, especially because it provides an upper bound on the L1 norm of the estimation error (averaged over algo randomness), rather than the expected estimation error commonly studied in the literature.  However, as pointed out below, the real usefulness of the L2 stability which introduced in the paper might be a bit questionable.

The results presented in Section 3 for the particular class of meta-algorithms are also interesting and useful. By the way, are the results presented in Sections 3/4 the first ones on the generalization of that type of algorithms?

**Weaknesses:**

The L2-type notion stability may be considered "new", although it is a reasonable and natural "norm-Jensenized" generalization of (2). In terms of contribution, this is probably sufficient, although it may also be considered derivative relative to prior works, at least conceptually.

All the results presented are for boosted algorithms, which is interesting on the one hand, but it is unclear if the results and the respective techniques produce non-trivial or improved results for standard algorithms, such as (mini-batch) SGD. Maybe the authors would like to elaborate more on this. Also see my comment about Remark 3, below.

Also, the results in Section 4 are quite weak. In particular, although the bounds are indeed decaying, the decays are extremely slow, like $\ log{(N)} / \sqrt{N}$, which is almost constant (so to say). I therefore doubt that such bounds could be of actual interest, apart from strictly theoretical.

**Questions:**

Line 126: What "sharper" means here? The bound is the same but with different constants, and with the "Jensen" version of the estimation error.

In Remark 3, although I understand the point, I believe that the word "tighter" is not really applicable, because of the different type of stabilities in Theorem 1 and in (4). Maybe you could rephrase. I see, however, that in the appendix the L2 stability of SGD (for instance) is established. I think that the relevant result could be highlighted more explicitly, even if Theorem 1 is in-expectation relative to \xi.

Title of Section 4 is misleading, and all algorithms considered are boosted. I would suggest that the authors rephrase, and also state more clearly throughout the paper that the bounds derived for specific algorithms are not for standard schemes such as SGD, but for the boosted versions.

It seems that Remark 6 has a technical issue: If I am not mistaken, the rate of Hardt et al. in the smooth , Lipschitz, convex case is of order $\sqrt{T}/N$ for a stepsize equal to $2/(L\sqrt{T})$, that is for a constant stepsize. In contrast, Remark 6 refers to a decaying root stepsize, which when summed from 1 to T gives a subharmonic number which is strictly larger that $\sqrt{T}$. If I am correct in the above, please make an effort to revise the results in Section 4 accordingly. See also lines 141-147, where a similar statement is made.



**Limitations:**

Yes

---

> ### Author Rebuttal · Authors · 2023-08-09
>
> Thank you for your insightful review and encouranging comments.
>
> > **Your comment:**  By the way, are the results presented in Sections 3/4 the first ones on the generalization of that type of algorithms?
>
> **Our response:** Yes, we for the first time establish in Theorem 2 a generalization bound for confidence-boosted randomized algorithms with $L_2$-uniform stability, which has not been possible to get under the weaker notion of on-average stability. The implications of Theorem 2 for the boosted SGD with decaying learning rates are also new to our knowledge.
>
>
> > **Your comment:** All the results presented are for boosted algorithms, which is interesting on the one hand, but it is unclear if the results and the respective techniques produce non-trivial or improved results for standard algorithms, such as (mini-batch) SGD. Maybe the authors would like to elaborate more on this.
>
> **Our response:** Regarding the exponential generalization bounds for the original randomized algorithms with $L_2$-uniform stability, it is a thrilling question yet so far our confidence-boosting technique does not allow us to resolve this open problem completely. We believe that developing tighter martingale concentration inequalities for randomized algorithms with conditional uniform stability (conditioned on the algorithm random bits), if possible, would be a vital step towards fully resolving this challenging problem.
>
>
> >**Your comment:** In particular, although the bounds are indeed decaying, the decays are extremely slow, like
> $\log(N)/\sqrt{N}$, which is almost constant (so to say).
>
> **Our response:** We would like to highlight that in view of the lower bound of Bousquet et al. [2020, Proposition 9], up to logrithmic factors, the $\log(N)/\sqrt{N}$ rate is actually almost the best we can hope in general for *high-probability* generalization bounds.
>
>
> >**Your comment:** Line 126: What "sharper" means here? The bound is the same but with different constants, and with the "Jensen" version of the estimation error.
>
> >In Remark 3, although I understand the point, I believe that the word "tighter" is not really applicable, because of the different type of stabilities in Theorem 1 and in (4). Maybe you could rephrase.
>
> **Our response:** By saying that our result is sharper, what we really want to express is that our first-moment generalization bound in Theorem 1 implies the on-average bound in Equation (4), though they are obtained under different notions of stability while the rates are identical. We will rephrase the statement to more accurately reflect the strength of our results in future version.
>
>
> > **Your comment:** I see, however, that in the appendix the L2 stability of SGD (for instance) is established. I think that the relevant result could be highlighted more explicitly, even if Theorem 1 is in-expectation relative to \xi.
>
>
> **Our response:** We appreciate your suggestion about highlighting the implications of Theorem 1 for vanilla SGD with decaying learning rates, in addition to those of Theorem 2 for boosted SGD.
>
>
> > **Your comment:** Title of Section 4 is misleading, and all algorithms considered are boosted. I would suggest that the authors rephrase, and also state more clearly throughout the paper that the bounds derived for specific algorithms are not for standard schemes such as SGD, but for the boosted versions.
>
> **Our response:** Per your suggestion, we will modify the title of Section 4 and rephrase the texts to more accurately state that the results are custermized for confidence-boosted versions of SGD.
>
>
> > **Your comment:** In contrast, Remark 6 refers to a decaying root stepsize, which when summed from 1 to T gives a subharmonic number which is strictly larger that $\sqrt{T}$.
>
> **Our response:** There might be a misunderstanding with regard to this technical issue. Indeed, under the considered $\mathcal{O}(\frac{1}{\sqrt{t}})$ stepsizes in Remark 6, the rate of Hardt et al. [2016] is still of scale $\frac{\sqrt{T}}{N}$ due to the fact $\sum_{t=1}^T \frac{1}{\sqrt{t}} \le 2\sqrt{T}$.
>
>
>
> ## References:
>
> Olivier Bousquet, Yegor Klochkov, and Nikita Zhivotovskiy. Sharper bounds for uniformly stable
> algorithms. *COLT*, pp. 610–626, 2020.
>
> Moritz Hardt, Ben Recht, and Yoram Singer. Train faster, generalize better: Stability of stochastic
> 373 gradient descent. *ICML*, 1225–1234, 2016.

---

> > ### Comment · Reviewer_RTFd · 2023-08-16
> >
> > I would like to thank the authors for providing satisfactory responses to my comments, and for the very detailed clarifications to some of my technical questions (for instance, the log(N)/N issue is now fully clarified -though somewhat disappointing, but as it turns out this is how things are...).
> >
> > Please try to incorporate as many details as possible and/or amend accordingly the points in the text, in the revised version of the paper.
> >
> > I will happily increase my score to a 6.

---

> > > ### Author Response · Authors · 2023-08-17
> > > **Thank you for your feedback**
> > >
> > > We sincerely thank the reviewer for upgrading the score and providing very helpful comments for improvement.

---

### Official Review · Reviewer_Yv2V · 2023-07-07

**Soundness:** 3 good
**Presentation:** 3 good
**Contribution:** 2 fair
**Rating:** 7
**Confidence:** 4

**Summary:**

Stability is a useful tool for obtaining sharper generalization bounds in learning theory. In prior work, the notion of "on-average" stability is as follows. A randomized learning algorithm is stable if for all datasets $S$, adjacent datasets $S'$ (i.e. differing by a single point), and all data points $Z$, the expected difference (taken over the randomness of the algorithm) between the loss incurred by the algorithm at Z when trained on $S$ and $S'$ is bounded.

On-average stability can be used to provide "on-average" risk bounds, where the population risk and the empirical risk of the trained classifier have low expected difference. Note here that this does not guarantee that the empirical risk is close to the population risk -- it rather guarantees that the empirical risk is close to an unbiased estimator of the population risk. Specifically, the quantity $$|\mathbb{E}_{\xi}[R(A(S), \xi) - R_n(A(S), \xi)]|,$$ is bounded, where the absolute values occur outside the expectation.

The goal of this is work to, under tighter assumptions, create tighter risk estimation bounds where we can instead bound the absolute difference between the two risks (i.e. proving accuracy) rather than simply providing a "close-to-unbiased" estimator.

To do so, they begin with a tighter notion of stability, where they bound the expected squared difference between the loss incurred at Z when trained on $S$ and $S'$. By simply adding a square here, this eliminates the possibility of "large differences" cancelling each other out when the expectation taken, and thus is a much stronger notion of stability. Under this assumption, their main result is a risk estimation bound on $$\mathbb{E}_{\xi}|R(A(S), \xi) - R_n(A(S), \xi)|.$$ Here the absolute value appears inside the expectation, and this is the crucial difference.

**Strengths:**

This paper appears to be a significant technical development over prior work. While the notion of stability has been known to lead to better generalization bounds, this paper greatly strengthens the type of generalization bounds that can be obtained for _randomized_ learning algorithms (my understanding is that this paper offers no new insight into deterministic algorithms as their bound essentially reduces to existing bounds).

**Weaknesses:**

I think that the subtly of the contribution of this paper needs to be spelled out more clearly. As far as I can tell, the crucial difference between this bound and prior bounds is literally upon the placement of the absolute value signs. The quantities $|\mathbb{E}[R-R_n]|$ and $\mathbb{E}|R-R_n|$ are substantially different meaning that this difference is important, but nevertheless they visually look very similar to the reader. If my understanding of this paper is correct, then highlighting this difference could be helpful.

**Questions:**

Can you argue more explicitly why your bound is "sharper than its on-average counterpart?" I had to read through the algebra for this to become clear to me, and I generally don't like relying on algebra alone due to its sensitivity to typos.

Edit: these concerns have been addressed in the rebuttal.

**Limitations:**

None.

---

> ### Author Rebuttal · Authors · 2023-08-09
>
> Thank you for your insightful review and positive evaluation of our work.
>
> > **Your comment:** I think that the subtly of the contribution of this paper needs to be spelled out more clearly.
>
> **Our response:** Thanks for your concise summary of the crucial differences between our first-moment generalization bound (under $L_2$-stability) and the prior on-average generalization bounds (under on-average stability). Per your suggestion, we will more explicitly highlight the mentioned subtle differences in Section 1.2 and Section 2.4.
>
> > **Your comment:** Can you argue more explicitly why your bound is "sharper than its on-average counterpart?" I had to read through the algebra for this to become clear to me, and I generally don't like relying on algebra alone due to its sensitivity to typos.
>
> **Our response:**  While our bound in Theorem 1 has identical convergence rate to that of the on-average bound in Equation (4), the former is stronger in the sense that the expectation is taken outside the absolute generalization gap and thus implies the latter. We will more precisely elaborate on this difference in the updated draft.
>
> > **Minor comments** on typos.
>
> **Our response:** Thanks for catching the typos!

---

> > ### Comment · Reviewer_Yv2V · 2023-08-19
> > **Response to Rebuttal**
> >
> > Thanks for your response. It appears that my understanding of the paper was correct, and I think with added clarity this will be a strong paper that should be accepted. I'm consequently increasing my score and confidence.

---

> > > ### Author Response · Authors · 2023-08-20
> > > **Thank you for your response**
> > >
> > > We sincerely thank the reviewer for kindly increasing the rating score and the confidence of evaluationa as well. We will follow your advices to further improve the clarity of presentation in the revised paper.

---

### Official Review · Reviewer_ZVGG · 2023-07-10

**Soundness:** 4 excellent
**Presentation:** 4 excellent
**Contribution:** 3 good
**Rating:** 8
**Confidence:** 3

**Summary:**

Algorithmic stability refers to the sensitivity of learning algorithms to the training set. Roughly speaking, if a learning algorithm is more stable, then its generalization gap (i.e., its training loss is highly indicative of its population loss) can be better controlled. In the literature, various notions of algorithmic stability have been proposed for *randomized* learning algorithms. Two most prominent ones are *on-average uniform stability*, which says that the averaged (over its internal randomness) algorithm is uniformly stable, and *high probability uniform stability,* which says that the algorithm is uniformly stable with high probability over its internal randomness.

This paper puts forth a new notion of algorithmic stability called *L2-uniform stability*, which can be seen as a second moment version of on-average stability. This notion sits naturally between on-average stability, which is rather weak and thus leads to weaker generalization bounds, and high probability stability, which can be too stringent to satisfy. To demonstrate the utility of this novel notion, the authors show that L2-stability can be used to derive generalization bounds with (exponentially) high confidence, where the randomness is over *both* the training samples and the algorithm’s internal randomness. This is in contrast to the generalization bound derived from on-average stability which only holds *in expectation* with respect to the algorithm’s internal randomness. The authors provide applications of their novel technique by showing qualitatively stronger generalization bounds for SGD with respect to convex loss functions.

**Strengths:**

- **Simple and novel notion with useful applications.** L2-uniform stability is a natural strengthening of on-average uniform stability. Though the modification may seem minor, this novel notion leads to significantly stronger generalization bounds. As mentioned previously, a key deficiency of generalization bounds from on-average stability is that they only hold *in expectation* with respect to the algorithm's randomness. Certainly, generalization bounds that hold *with high probability* over the algorithm's randomness (and also randomness of training samples) are preferable. The first-moment generalization bound from L2-stability, combined with bagging [Breiman, 1996], yields such high probability bounds. The authors further show that SGD in various settings (e.g., for smooth/non-smooth convex losses) is sufficiently L2-stable, leading to sharper generalization bounds.

- **Clear exposition.** The ideas in the paper are presented with great clarity and the overall text, including the technical proof sections, flows seamlessly.

**Weaknesses:**

N/A

**Questions:**

- Is it possible, perhaps under additional assumptions, to get exponentially high probability generalization bounds from on-average uniform stability? What are some technical barriers that stand in the way of boosting the confidence of generalization bounds from on-average stability?

**Minor comments**
- Typo in line 56: "deviatin"

**Limitations:**

Yes.

---

> ### Author Rebuttal · Authors · 2023-08-09
>
> Thank you for your insightful review and appreciation of our work.
>
> > **Your comment:** Is it possible, perhaps under additional assumptions, to get exponentially high probability generalization bounds from on-average uniform stability? What are some technical barriers that stand in the way of boosting the confidence of generalization bounds from on-average stability?
>
> **Our response:** Thanks for the interesting questions which we hope can be addressed by the following replies:
>
> 1. In general, it still remains open, if not impossible, to derive near-tight exponential generalizations bounds via confidence-boosting technique under the notion of on-average uniform stability. A substantial challenge faced here is that from on-average stability, it seems only possible to get exponetial generalization bounds w.r.t the on-average loss like in Equation (4), to which the confidence-boosting trick is not directly applicable.
>
> 2. It is however possible to get exponential bounds on *excess risk* under the weaker notion of on-average uniform stability. Indeed, as discussed in the last paragrah of Section 3 that the exponential excess risk bound in Equation (14) can be obtained by applying the confidence-boosting technique to the in-expectation excess risk bound in Equation (13), keeping in mind that excess risk is always non-negative.
>
> 3. If we are willing to modify the confidence-boosting procedure using *subbagging*, i.e., we repeat the algorithm on independent subsets and take the best one on the holdout, then it is indeed possible obtain the desirable near-optimal generalization bounds under on-average stability. However, such a kind of subbagging procedure needs to run the randomized algorithm over $\log(1/\delta)$ independent subsets instead of the full dataset. In contrast, our bagging-style strategy repeatedly executes the randomized algorithm over the entire sample (except the holdout) with $\log(1/\delta)$ independent random seeds, which is expected to be more favorable for practical implementation.

---

> > ### Comment · Reviewer_ZVGG · 2023-08-14
> >
> > Thank you for addressing my questions and providing further insight!

---

> > > ### Author Response · Authors · 2023-08-15
> > > **Thank you for your acknowledgement**
> > >
> > > Many thanks for your kind feedback!

---

### Official Review · Reviewer_3pL8 · 2023-07-11

**Soundness:** 3 good
**Presentation:** 3 good
**Contribution:** 3 good
**Rating:** 7
**Confidence:** 3

**Summary:**

A new notion of algorithmic stability is introduced for randomized algorithms, which is intermediate to previously studied notions and leads to new strong generalization bounds. The need for a new notion is justified due to known lower bounds under previously studied notions by related to on-average stability, and practical limitations of stronger notions like uniform stability with high probability when applied to SGD. A new intermediate L2-uniform stability is introduced with a goal to fill this gap, and generalization guarantees are obtained under this notion improving over on-average stability bounds. Also boosting is used to give high probability generalization error bounds. Remarkably, the stability of SGD is bounded under this new notion for convex and/or smooth loss functions and the stability constants are derived.

**Strengths:**

- The paper considers a novel and interesting notion of stability, and establishes strong generalization bounds for algorithms that are stable under the defined sense of stability.
- High probability risk bounds are obtained via boosting.
- The proposed notion of stability is shown to be satisfied for the popular SGD (stochastic gradient descent) algorithm when the loss function is convex and/or smooth.

**Weaknesses:**

- No lower bounds are provided, so tightness of the results is not clear. The upper bounds are tighter, but under a stronger assumption than previously studied in Bousquet et al. 2020.
- The presentation of Section 4 could be improved. What are the novel insights in establishing the bounds on the L2-uniform stability parameters? For implications to SGD, it might be better to present/discuss the lemmas on the L2-uniform stability parameters in the main body, possibly replacing the corresponding corollaries obtained by combination with Theorem 2.

**Questions:**

- How does $\gamma_{L_2,N}$ compare to $\gamma_N$ for SGD?
- Line 75: Should it be $[0,M]$ instead of $(0,M]$?
- Line 187: Typo: McDiamid’s

**Limitations:**

Limitations of results could be discussed in more detail. For example, in Remark 6, are there terms where the proposed bounds are worse than the in-expectation bounds of Hardt et al. 2016?

---

> ### Author Rebuttal · Authors · 2023-08-09
>
> Thank you for your insightful review and appreciation of our work.
>
> > **Your comment:** No lower bounds are provided, so tightness of the results is not clear.
> >
> **Our response:** Per your comment, we plan to provide some additional analysis regarding tightness of our bounds in the next version. Since the main result of Theorem 1 is implied by the the moment bound for sums of randomized functions in Proposition 2, it is natural to ask if that latter bound can be improved in general. Inspired by the lower bound analysis of Bousquet et al. [2020, Proposition 9], we indeed believe that similar results and analysis could be extended to sum of randomized functions of Rademacher signs that satisfy the L2-bounded-difference property. A potential way to construt such randomized functions is to consider $g_i(S,\xi)=MZ_i + \frac{\beta}{2}Z_i\left(\sum_{j\neq i}Z_j + \xi_1\right)\xi_2$ where $Z_i$, $\xi_1$, $\xi_2$ are all Rademacher signs. It can be verified that these functions have L2-bounded-difference property, and the arguments of Bousquet et al. [2020, Proposition 9] can be naturally adapted to the sum of $g_i(S,\xi)$ to get the desired lower bound that matches the upper bound presented in Proposition 2.
>
> > **Your comment:** The presentation of Section 4 could be improved. What are the novel insights in establishing the bounds on the L2-uniform stability parameters?
>
> **Our response:**  We appreciate your suggestions about improving the exposition of results in Section 4. To compare with the on-average uniform stability analysis of SGD by Hardt et al. [2016], a non-trivial technical contribution of our L2-uniform stability analysis is using Lemma 10 to handle a key recurssion appeared in Equation (32) for smooth and convex losses.
>
>
> > **Your comment:** How does $\gamma_{L_2,N}$ compare to $\gamma_N$ for SGD?
>
> **Our response:** In the following, we compare $\gamma_{L_2,N}$ to $\gamma_N$ for SGD with smooth or non-smooth convex losses:
>
> 1. For smooth and convex losses, $\gamma_{L_2,N}\le \sqrt{\frac{1}{N}\sum_{t=1}^T \eta^2_t + \frac{1}{N^2} \left(\sum_{t=1}^T \eta_t\right)^2}$ (see Lemma 12). In comparson, $\gamma_N$ is known to be of the scale $\frac{1}{N}\sum_{t=1}^T\eta_t$ [Hardt et al., 2016, Theorem 3.7].
>
> 2. For non-smooth and convex losses, $\gamma_{L_2,N}\le \sqrt{\sum_{t=1}^{T}\eta^2_t + \frac{1}{N^2} \left(\sum_{t=1}^T \eta_t\right)^2}$ (see Lemma 13), while $\gamma_N$ scales similarly as $\sqrt{\sum_{t=1}^{T}\eta^2_t} + \frac{1}{N}\sum_{t=1}^T\eta_t$ accroding to Bassily et al. [2020, Theorem 3.2].
>
>
> > **Your comment:** Line 75: Should it be $[0,M]$ instead of $(0,M]$? Line 187: Typo: McDiamid’s
>
> **Our response:** Yes, the range of loss value should be $[0,M]$. Thanks for catching the typos!
>
>
> ## References:
>
> Olivier Bousquet, Yegor Klochkov, and Nikita Zhivotovskiy. Sharper bounds for uniformly stable
> algorithms. *COLT*, pp. 610–626, 2020.
>
> Moritz Hardt, Ben Recht, and Yoram Singer. Train faster, generalize better: Stability of stochastic
> 373 gradient descent. *ICML*, 1225–1234, 2016.
>
> Raef Bassily, Vitaly Feldman, Cristobal Guzman, and Kunal Talwar. Stability of stochastic gradient descent on nonsmooth convex losses. *NeurIPS*, 1–10, 2020.

---

> > ### Comment · Reviewer_3pL8 · 2023-08-13
> >
> > Thanks for the response and clarification for my questions.
> >
> > The lower bound analysis looks promising. In particular, it would be good to highlight the differences relative to Bousquet et al. (2020) since their analysis seems to be tight up to logarithmic terms.

---

> > > ### Author Response · Authors · 2023-08-13
> > > **Thank you for your acknowledgement**
> > >
> > > We appreciate your prompt feedback and are glad to know that our response addressed your concerns raised in the initial review. Thanks for the suggestion about highlighting the differences to the work of Bousquet et al. [2020], which we will discuss in more details in the next version.

---

### Decision · Program_Chairs · 2023-09-21

**Decision:**

Accept (poster)

**Comment:**

This meta review is based on the reviews, the authors rebuttal and the discussions with the reviewers, discussions with the SAC, and ultimately my own judgement on the paper. There was a consensus that the paper contributes sound and interesting contributions to generalization theory. I feel this work deserves to be featured at NeurIPS and will attract interest from the community. I would like to personally invite the authors to carefully revise their manuscript to take into account the remarks and suggestions made by reviewers in their camera-ready. Congratulations!